# Targeted Unlearning Using Perturbed Sign Gradient Methods With Applications On Medical Images

**George R. Nahass** *gnahas2@uic.edu*
*Department of Biomedical Engineering*
*Department of Ophthalmology*
*University of Illinois Chicago*

**Zhu Wang**
*Department of Computer Science*
*University of Illinois Chicago*

**Homa Rashidisabet**
*Department of Ophthalmology*
*University of Illinois Chicago*

**Won Hwa Kim**
*Computer Science and Engineering*
*Pohang University of Science and Technology, South Korea*

**Sasha Hubschman**
*Department of Ophthalmology*
*University of Illinois Chicago*

**Jeffrey C. Peterson**
*Department of Ophthalmology*
*University of Illinois Chicago*

**Pete Setabutr**
*Department of Ophthalmology*
*University of Illinois Chicago*

**Chad A. Purnell**
*Department of Plastic and Reconstructive Surgery*
*University of Illinois Chicago*

**Ann Q. Tran**
*Department of Ophthalmology*
*University of Illinois Chicago*

**Darvin Yi**
*Department of Biomedical Engineering*
*Department of Ophthalmology*
*University of Illinois Chicago*

**Sathya N. Ravi** *sathya@uic.edu*
*Department of Computer Science*
*University of Illinois Chicago*

**Reviewed on OpenReview:** *https://openreview.net/forum?id=XEObJg6sQN*

## Abstract

Machine unlearning aims to remove the influence of specific training samples from a trained model without full retraining. While prior work has largely focused on privacy-motivated settings, we recast unlearning as a general-purpose tool for post-deployment model revision. Specifically, we focus on utilizing unlearning in clinical contexts where data shifts, device deprecation, and policy changes are common. To this end, we propose a bilevel optimization formulation of boundary-based unlearning that can be solved using iterative algorithms. We provide convergence guarantees when first order algorithms are used to unlearn and introduce a tunable loss design for controlling the forgetting–retention tradeoff. Across benchmark and real-world clinical imaging datasets, our approach outperforms baselines on both forgetting and retention metrics, including scenarios involving imaging devices and anatomical outliers. This work demonstrates the feasibility of unlearning on clinical imaging datasets and proposes it as a tool for model maintenance in scenarios that require removing the influence of specific data points without full model retraining. Code is available here.

## 1    Introduction

In recent years, the awareness of the public regarding data ownership has continued to increase. As large deep learning models continue to be trained using information from people who may not have explicitly opted into such a procedure, the question naturally arises: how can they 'opt-out' post-fact? To this point, in 2016, the General Data Protect Regulation (GDPR) was passed by the EU, stating that individuals have the 'right to be forgotten' and businesses have the 'obligation to erase personal data' Regulation (2020); Graves et al. (2021). As the cost and training time continue to increase, if an individual decides to exercise their right to be forgotten, retraining the model from scratch without their data may prove to be infeasible from both a logistic and economic point of view Cottier et al. (2024). Machine unlearning has emerged as a primary strategy for such purposes Liu et al. (2024a;b).

While machine unlearning was originally developed to satisfy privacy mandates, its potential applications extend well beyond regulatory compliance Kurmanji et al. (2024). In real-world machine learning pipelines, particularly in healthcare, there may be a need to revise models over time due to shifting data distributions, evolving clinical protocols, or the gradual deprecation of imaging devices Guo et al. (2021); Moreno-Torres et al. (2012); Futoma et al. (2020). Continual learning has been proposed as a method to continuously update models, but this does not allow for removal of specific samples which may be necessary in various settings Challen et al. (2019); Lee & Lee (2020). In these scenarios, retraining from scratch is often infeasible due to computational cost, regulatory restrictions on data reuse, or lack of access to the full original dataset De Lange et al. (2021); Verwimp et al. (2023). However, very few studies have explored unlearning as a practical mechanism for model maintenance and revision in real world settings.

Despite growing interest in machine unlearning, significant limitations remain in both experimental scope and methodological flexibility. Most prior work focuses on benchmark datasets such as CIFAR-10 and MNIST, which lack the resolution, heterogeneity, and domain-specific information of real-world datasets Greenspan et al. (2016); Litjens et al. (2017). Furthermore, selective unlearning, in which only specific subsets of data (e.g., based on metadata or image quality) are removed, is rarely explored. Here, fewer methods provide meaningful control over the forgetting–retention tradeoff, limiting their utility in settings where both privacy and performance must be carefully balanced Kurmanji et al. (2024).

In this work, we extend the boundary-based unlearning framework of  Chen et al. (2023) by recasting it as a bilevel optimization problem that can be solved using first-order methods. Our formulation enables a principled, modular, and scalable approach to deep model unlearning, offering both formal convergence guarantees and empirical improvements across multiple datasets. Our framework supports both sample-level and subgroup-level unlearning, reflecting real-world deployment scenarios where full metadata is often unavailable at test time, but subgroup-based revisions (e.g., removing the influence of outdated scanners or low-quality images) are required. To our knowledge, this is the first general-purpose unlearning framework applied to clinical imaging datasets, and the first to introduce an algorithm for both instance-level and distributional-level model maintenance.

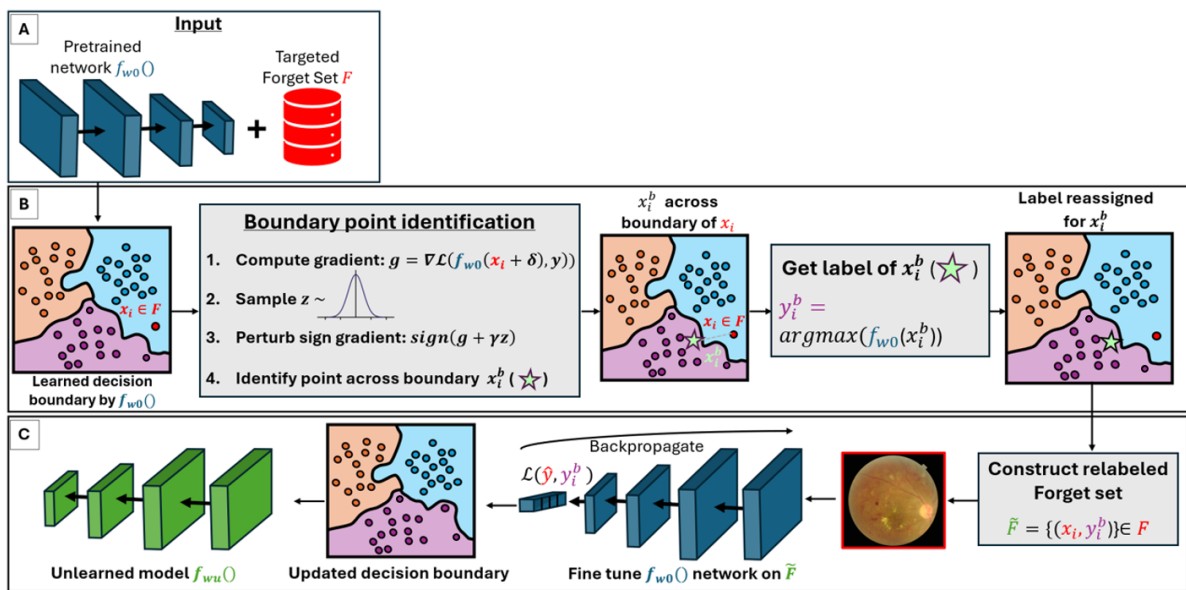

Figure 1: Graphical schematic of our proposed unlearning algorithm. We begin with a pretrained CNN $f_{w_0}$ and a user-defined forget set $F$ (A). In the inner optimization loop (B), we identify boundary points $x_i^b$ across the decision surface of the original model via our perturbed sign-gradient method. For each forget sample $x_i$, we assign a new label $y_i^b = \arg\max f_{w_0}(x_i^b)$ based on the closest incorrect class, and construct a relabeled forget set $\tilde{F} = \{(x_i, y_i^b)\}$. In the outer optimization loop (C), we fine-tune the model on $\tilde{F}$, optionally incorporating remain-set supervision. The result is an unlearned model $f_{w_u}$ whose decision boundaries are shifted to forget the designated samples.

**Related Work.** To contextualize our contributions, we briefly review existing propositions for unlearning deep neural networks through parameter modification. In Golatkar et al. (2020), the Fisher Information Matrix (FIM) is used to dynamically compute the appropriate noise to scrub the parameters of a Deep Neural Network of the Forget set. This idea requires approximation of the Hessian through the FIM, which may be expensive to compute. To alleviate this, Peste et al. (2021) uses gradients of samples to be unlearned to approximate the FIM for faster unlearning. In a similar vein, Mehta et al. (2022) approximate the Hessian using L-FOCI to choose a subset of parameters for unlearning. Chen et al. (2023) introduced a simple decision-boundary method achieving SOTA, but lacks convergence guarantees and offers limited flexibility in balancing forgetting and retention. For a comprehensive review and in depth descriptions on various trends and techniques in machine unlearning, we point readers to Xu et al. (2023) and Nguyen et al. (2024).

## 2 Methods

### 2.1 Problem Setup

We consider supervised classification tasks with deep neural network models $f_w : \mathcal{X} \to \mathbb{R}^K$ parameterized by weights $w$, trained on a dataset $D = \{(x_i, y_i)\}_{i=1}^N$ of image–label pairs, where each $y_i \in \{e_1, \dots, e_K\}$ is a one-hot vector in $\{0, 1\}^K$ where $K$ is the number of classes.

Model $f_w$ is trained by minimizing a suitable loss function $\mathcal{L}$ such as cross entropy to predict labels $y_i$ given image $x_i$. After deployment, a subset $F \subset D$ is designated for removal due to privacy, policy, or maintenance considerations. We use $R = D \setminus F$ to denote the "remain" set that are available for the task. The machine unlearning problem asks: how can we update $f_w$ to a new model $f_{w_u}$ that forgets $F$ while retaining performance on $R$, without retraining from scratch with $R$?

Here, we define unlearning success via two desiderata on model $f_w$: **(C1) Forgetting:** the updated model should misclassify $(x, y) \in F$, i.e., $f_{w_u}(x) \neq y$; **(C2) Remaining:** the updated model should preserve accuracy on $R$, i.e., $f_{w_u}(x) = y$ for all $(x, y) \in R$.

In practice, we evaluate these goals via held-out splits of $F$ and $R$. To formalize our algorithm, we view selective unlearning as a bilevel optimization problem that seeks to modify the model's decision boundaries in a targeted and controlled manner.

We note that our formulation of desideratum (C1), which requires misclassification of the forget set, slightly differs from the standard definition of machine unlearning, which typically evaluates success by the statistical indistinguishability of the unlearned model from one retrained on $R$ Golatkar et al. (2020); Kurmanji et al. (2024); Xu et al. (2023). In privacy-driven applications, the retraining paradigm is appropriate, as if an individual requests removal, the primary objective is to mimic the distribution of a model trained without their data without necessarily forcing any particular behavior on $F$. Our work, however, is motivated by model maintenance, where the goal is not only to ensure removal of outdated or undesirable data, but also to prevent such data from continuing to influence predictions in deployment. In this setting, requiring misclassification provides an operational guarantee: samples in $F$ are actively pushed away from their original labels, ensuring that they cannot contribute useful signals to future predictions. See Section 6 for implications of this method in medical imaging.

Additionally, prior work often considers resistance to membership inference attacks (MIA) as an additional desideratum for unlearning in privacy-focused applications. However, as our emphasis is on clinical model maintenance we treat MIA as an important evaluation metric rather than a core desideratum and report it alongside C1 and C2 for completeness and comparison.

## 2.2 Unlearning as Bilevel Optimization

Because post-hoc unlearning must modify a learned model without full retraining, recent work has focused on shifting its decision boundaries by modifying parameters $w$ Chen et al. (2023). We build on this framework which proposes forgetting a sample by modifying the model's decision boundary through a two-phase process. First, for each forget set sample $(x, y) \in F$, the method computes a nearby point $x^b$ across the decision boundary. Second, this point is used to assign a new label $y^b \neq y$ for $x$, and the model is fine-tuned on $(x, y^b)$ to induce forgetting.

We unify both steps into a single bilevel optimization framework:

$$
\begin{aligned}
\min_{w} \quad & \sum_{i \in R} \mathcal{L}(f_w(x_i), y_i) - \sum_{i \in F} \mathcal{L}(f_w(x_i), f_{w_0}(x_i^b)) \\
\text{s.t.} \quad & x_i^b - x_i \in \arg\min_{\delta : f_{w_0}(x_i + \delta)^\top y_i \leq \kappa} \ell(\delta) \quad \forall i \in F,
\end{aligned}
\tag{1}
$$

where $w_0$ are the original model weights, $\mathcal{L}$ is the supervised loss (e.g., cross-entropy), $\ell(\cdot)$ is an inner loss function designed to penalize large perturbations (such as $\ell_p$ norms), and $\kappa < 1/K$ ensures the boundary point $x_i^b$ lies across the decision surface of class $y_i$. Choosing $\kappa = 1/K$ corresponds to projecting $x_i$ exactly onto the decision boundary, whereas lesser values correspond to finding $x_i^b$ across the boundary. See Appendix Section A.6 for more details.

This objective jointly optimizes the new model weights $w$ while enforcing that forget samples receive mismatched predictions. The inner constraint defines the boundary search: for each forget sample $x_i$, we identify the minimal perturbation $\delta$ that crosses the decision boundary, subject to a proximity constraint via $\ell(\delta)$. The resulting boundary point $x_i^b = x_i + \delta$ is used to guide the outer update. A visualization of our entire algorithm can be seen in Figure 1.

It should be noted that the second term in Eq. 1 coincides with the third term in Eq. 3 of Kurmanji et al. (2024), enabling our formulation to be viewed as a flexible student–teacher framework. For instance, one may steer unlearning toward specific incorrect classes by modifying the boundary label assignment $y^b$.

*Remark* 2.1. (On the definition of forgetting.) We note that our formulation of desideratum (C1), which requires misclassification of the forget set $F$, differs slightly from the standard definition of machine unlearning.

Unlearning success is typically defined by statistical indistinguishability between the unlearned model and a model retrained from scratch on the retain set $R$ Golatkar et al. (2020); Kurmanji et al. (2024); Xu et al. (2023). Under that paradigm, accuracy on $F$ is expected to resemble the retrain baseline, which is often close to random chance for balanced datasets but not necessarily zero. In contrast, our operationalization reflects the needs of model maintenance, where random like predictions, as required by the standard definition, is not necessary.

We present an equivalent unconstrained formulation of inner maximization in (1) that can be used to relate the Boundary Shrink method in Chen et al. (2023):

*Lemma* 2.2 (Unconstrained Unlearning). *Assume $(x_i, y_i) \in F$ with $y_i$ to be 1-hot vector representation $x_i$'s class label, and $\mathcal{L}(\cdot, \cdot)$ is a smooth function that decomposes with respect to coordinates of $y_i$, and is decreasing. The inner minimization problem in* (1) *is equivalent to* $\max_\delta \mathcal{L}(f(w_0, \delta + x_i), y_i)$. $f(w_0, \delta + x_i)$ *denotes the label predicted by model with parameters $w_0$ for the perturbed sample $\delta + x_i$.*

The proof of Lemma 2.2 is in Appendix A.1. In essence, we apply Karush-Kuhn-Tucker (KKT) conditions Nocedal & Wright (2006) on the inner minimization problem. The 1-hot representation assumption of $y$ in Lemma 2.2 is not necessary – our result can be extended to multilabel classification tasks also. Please see Appendix for more details. Moreover, we show that the above Lemma 2.2 is true for loss function such as Squared $\ell_2$ norm that are used for regression tasks in the Appendix.

*Corollary* 2.3 (Boundary Shrink Initialization). *Assume $\delta = 0$ at initialization. Then, maximizing $\mathcal{L}(f(w_0, \delta + x_i), y_i)$ with respect to $\delta$ using any first order method is equivalent to Boundary Shrinking.*

The proof of Corollary 2.3 is in Appendix A.2, and involves comparing iterates using Chain rule. One main advantage of our formulation in (1) is that it is possible to specify a desired objective function (such as closest example in squared $\ell_2$ norm, $\ell_1$ for sparsity etc.) of $\delta$ to get minimum perturbation, if desired.

## 2.3 Solving the Inner Optimization Problem

The inner task in Eq. (1) is to find, for every forget sample $x_i$, a *nearby* point $x_i + \delta$ whose logit for the true class $y_i$ drops below the threshold $\kappa$. Because $f_{w_0}(x_i + \delta)$ is highly non-convex in $\delta$ Salman et al. (2019), the one-step FGSM update of Goodfellow et al. (2014) often stalls in poor local optima. We therefore adopt a *perturbed sign-gradient* scheme that enjoys provable convergence.

**Perturbed sign update.** We use Lemma 2.2 to reformulate the inner optimization constraint as a loss maximization. We refer to this loss maximization as "inner" loop from now on. The key insight is that this allows us to borrow techniques from adversarial attack literature as explained below. Let $g = \nabla_\delta \mathcal{L}\big(f_{w_0}(x_i + \delta), y_i\big)$ and draw $z \sim \mathcal{N}(0, I)$. With a decaying step size $\epsilon_t = c/t$ ($c > 0$) we calculate $d_t$ and update $\delta$ as,

$$d_t = \epsilon_t \, \text{sign}(g + \gamma z), \quad \delta \leftarrow \delta + d_t, \tag{2}$$

where the noise level $\gamma \geq 0$ is fixed for the inner loop. The Gaussian term, inspired by Langevin dynamics, helps the iterate escape sharp local minima while keeping the update aligned with the true gradient.

The next lemma (proved in Appendix A.3) shows that, in expectation, the update is ascent-aligned with the objective that drives the boundary search.

*Lemma* 2.4 (Ascent-direction guarantee). *With $d_t$ defined in Eq.* (2), $\mathbb{E}_z\big[d_t^\top g\big] \geq 0$, *with equality iff $g = 0$.*

**Convergence rate.** Since the update step is ascent-aligned in expectation with respect to the randomness of $z$, standard stochastic optimization argument can be used to provide convergence guarantees for our procedure as below.

*Theorem* 2.5 (Convergence of perturbed FGSM). *Assume we use update direction $d$ as in equation* (2) *for $T$ iterations with step size sequence $\epsilon_t = O(1/t), t = 1, ..., T$, and $\mathcal{L}$ has $L-$Lipschitz continuous gradient wrt $\delta$. Then the procedure converges to a point $\delta$ such that $\mathbb{E}_z\|\nabla_\delta \mathcal{L}\|_1 \leq \epsilon_{acc}$ in $T = O(DL/\epsilon_{acc})$ iterations where $D$ is the dimension of $\delta$.*

Please see (Appendix A.4) for our proof of Theorem 2.5.

---

**Algorithm 1:** Boundary Search via Perturbed Sign Gradient

---

**Input:** Pretrained model $f_{w_0}$, Forget set $F$, Step size schedule $\epsilon_t$, Perturbation scale $\gamma > 0$, Inner steps $T_{\text{inner}}$, (optional) closeness regularization parameter $\lambda > 0$

**Output:** Relabeled forget set $\tilde{F} = \{(x_i, y_i^b)\}$

Initialize $\tilde{F} \leftarrow \emptyset$;

**for** $(x_i, y_i) \in F$ **do**

    Initialize $\delta \leftarrow 0$;

    **for** $t = 1$ **to** $T_{inner}$ **do**

        Sample $z \sim \mathcal{N}(0, I)$;

        $g \leftarrow \nabla_\delta \mathcal{L}(f_{w_0}(x_i + \delta), y_i)$;

        $\delta \leftarrow \delta + \epsilon_t \cdot \text{sign}(g + \lambda \nabla_\delta \ell(\delta) + \gamma z)$;

    Set $x_i^b \leftarrow x_i + \delta$;

    Set $y_i^b \leftarrow \arg\max f_{w_0}(x_i^b)$;

    Store $(x_i, y_i^b)$ in $\tilde{F}$;

**return** $\tilde{F}$

---

**Optional closeness regularisation.** The above Theorem 2.5 guarantees that we find a point $x_i^b$ on the decision boundary. To find the closest boundary point to $x_i$, we may add the penalty $\ell(\delta) = \frac{1}{2}\|\delta\|_2^2$ weighted by a hyper-parameter $\lambda \geq 0$. The inner loop update then becomes,

$$d_t = \epsilon_t \, \text{sign}\big(g + \lambda \nabla_\delta \ell(\delta) + \gamma z\big). \tag{3}$$

Setting $\gamma = \lambda = 0$ **exactly** recovers the implementation of Boundary-Shrink update of Chen et al. (2023), demonstrating that their method is a special case of our framework. Our convergence analysis, however, applies specifically to the perturbed version of boundary shrinking presented here, rather than to this unperturbed special case. We treat $\lambda$ (along with $\gamma$ as hyperparameters which can be tuned. The solution of inner loop, whose output defines the boundary points $x_i^b = x_i + \delta$ which is used to guide the outer model update in Eq. 1. The pseudocode algorithm of phase 1 can be seen in Algorithm 1.

*Remark* 2.6. (Relation to Chen et al. (2023).) Both our approach and Chen et al. (2023) identify perturbed points near the decision boundary and fine-tune the model on relabeled samples. The main difference lies in how these boundary points are obtained and used. Chen et al. (2023) apply a single deterministic FGSM-style update with fixed step size, whereas we cast this step as an explicit inner optimization problem solved by a perturbed sign-gradient scheme with optional regularization and provable convergence. This generalization provides a theoretically grounded extension of the update while preserving the two-loop implementation.

## 2.4 Outer Optimization

Once the boundary point $x_i^b$ is identified for each forget sample $x_i \in F$, we solve the outer minimization to obtain updated model weights $w_u$. So the success of this two-phase bilevel optimization depends critically on identifying accurate boundary points $x_i^b$ in the inner loop. The goal in outer loop is to satisfy **(C1)**, as a large loss value $\mathcal{L}(f_w(x_i), y_i^b)$ is typically positively correlated with the distance between the model prediction and the reassigned label. This is particularly true in affine models of the form $f_w(x) = Wx + b$, where $\mathcal{L}$ is a monotonic function.

To assign the boundary label $y_i^b$, we select the class corresponding to the largest logit of the original model $f_{w_0}(x_i^b)$ as described in Algorithm 1. The new label is defined $y_i^b = \arg\max f_{w0}(x_i^b)$, i.e., the highest scoring non-true class at the boundary point. In principle, $y_i^b$ need not be the $\arg\max$ class and could instead be a specific target class towards which all forget samples are reassigned, thereby steering forgetting towards that class. The model is then fine-tuned on the reassigned forget set $\tilde{F} = ((x_i, y_i^b)_{i \in F}$ as:

$$w_u = \arg\min_w \sum_{i \in \tilde{F}} \mathcal{L}(f_w(x_i), y_i^b), \tag{4}$$

---

**Algorithm 2:** Model Update via Relabeled Forget Set

---

**Input:** Relabeled forget set $\tilde{F}$ from the inner loop (Alg. 1), Initial model weights $w_0$, Learning rate $\eta$,
       Outer steps $T_{\text{outer}}$
**Output:** Unlearned model weights $w_u$
Initialize $w \leftarrow w_0$;
**for** $t = 1$ **to** $T_{outer}$ **do**
    Sample minibatch $(x, y^b) \sim \tilde{F}$;
    $w \leftarrow w - \eta \cdot \nabla_w \mathcal{L}(f_w(x), y^b)$;
**return** $w_u \leftarrow w$

---

where $w_u$ are the updated weights after unlearning. The pseudocode of our vanilla outer loop can be seen in Algorithm 2.

**New Heuristics for Unlearning**  We additionally introduce three configurable extensions to the algorithm that enable controlled trade-offs between forgetting and retention: i) incorporating remain-set loss during the outer loop, ii) using top-k soft logit supervision to reduce irrelevant weight updates, and iii) enabling multi-objective tradeoffs by composing unlearned models across modules. Full mathematical details and pseudocode are provided in Appendix Section A.5. These are not additional hyperparameters but design choices within the optimization process, and their effects are systematically assessed in our ablation experiments (Section 4.3).

## 3 Experiments

### 3.1 Datasets

We evaluate our method on six image classification datasets: two standard benchmarks and four medical datasets. The benchmark datasets include CIFAR-10 and FashionMNIST Krizhevsky et al.; Xiao et al. (2017). The medical datasets include color fundus photographs (CFP-OS), magnetic resonance images (MRI) Nickparvar (2021), and two clinical datasets collected from real-world settings.

The CFP-OS dataset is a three-class dataset consisting of 1500 open-source images, curated from IDRID Porwal et al. (2018), ORIGA Zhang et al. (2010), REFUGE Pachade et al. (2020), and G1020 Bajwa et al. (2020). The MRI dataset includes 3000 images from each of four classes (Normal, Glioma, Meningioma, Pituitary), sampled from the dataset of Nickparvar (2021). Full details on the origin of the open-source datasets is provided in Appendix Table 3.

The first clinical dataset, which we call **CFP-Clinic**, consists of color fundus photographs drawn from the Illinois Ophthalmic Database Atlas (I-ODA) Mojab et al. (2021), a collection of over 3 million images collected over a 12-year period. We randomly sampled 1000 CFP images with ICD-10 codes corresponding to diabetic retinopathy (DR), glaucoma, or other retinal disorders (retinopathy of prematurity, retinal hemorrhage, and degenerative maculopathies). Each image belongs to one class, and no patient appears more than once.

The second clinical dataset, which we call **Oculoplastic**, consists of cropped periocular images from patients with oculoplastic conditions, spanning three categories: healthy, thyroid eye disease, and craniofacial dysmorphology. Healthy images were sampled from the Chicago Facial Dataset Ma et al. (2015). As with CFP-Clinic, all samples are from unique individuals. Metadata for both clinical datasets can be found in Figure 4. For medical imaging datasets (MRI, fundus, oculoplastic), no additional preprocessing was performed to prepare for unlearning compared to the general image datasets.

In clinical unlearning experiments, the forget/remain split is defined by the imaging device metadata associated with each sample in the I-ODA dataset or by periorbital distances involving the iris Nahass et al. (2025; 2024).

## 3.2 Model Training

For the medical image datasets (CFP-OS, MRI, CFP-Clinic, and Oculoplastic), we fine-tune a ResNet-50 pretrained on ImageNet-1k He et al. (2015), replacing the final classification layer with a dataset-specific output layer and applying a dropout of 0.5 prior to the final layer. For CIFAR-10 and FashionMNIST, we use the All-CNN architecture trained from scratch following Springenberg et al. (2015).

All training is performed using cross-entropy loss and SGD with momentum 0.9, a learning rate of $1 \times 10^{-3}$, and $L_2$ regularization with coefficient $1 \times 10^{-4}$. Data augmentation includes random horizontal flips, random rotations up to $15°$, color jittering, and random cropping. An 80/20 train–test split is used across all datasets. Training was terminated once the validation accuracy had converged (difference between training accuracy of previous epoch and current epoch is $< 1$, and the model with the highest validation accuracy within (latest) 5 epochs before convergence was chosen for testing purposes.

For medical datasets, the ResNet-50 was trained until convergence. For CIFAR-10 and FashionMNIST, the All-CNN is trained for 25 and 15 epochs, respectively, with a learning rate of 0.01 and batch size 64. All hyperparameters were selected via grid search. All experiments are run on across 3 NVIDIA GeForce RTX 2080 Ti GPUs.

## 3.3 Evaluation Metrics

We evaluate unlearning performance using four primary criteria. *Forget Set Accuracy (F-Acc)* measures the accuracy on a held-out portion of the forget set $F$; lower values indicate more effective forgetting. *Remain Set Accuracy (R-Acc)* captures the accuracy on a held-out portion of the remain set $R$; higher values reflect better retention of model utility. To assess the balance between these objectives, we compute the *F/R Accuracy Ratio*, defined as the ratio of forget to remain accuracy; lower values suggest better selective forgetting without compromising performance on $R$. Finally, we measure vulnerability to *Membership Inference Attacks (MIA) Carlini et al. (2022)* following the protocol of Kurmanji et al. (2024). A logistic regression classifier is trained to distinguish between samples in $F$ and a held-out test set drawn from the same class. The attack model takes as input the clipped loss $\mathcal{L}(f_w(x), y)$, truncated to the range $[-400, 400]$. We use 5-fold cross-validation and report average classification accuracy, where a perfect defense achieves 50%. Together, these metrics quantify whether forgetting was successful (**C1**), model utility was preserved (**C2**), and, even though not the main focus of our work, whether post-unlearning privacy risk was mitigated.

## 3.4 Baselines

We compare our method against several established baselines. *Retrain* serves as the gold standard: a model is retrained from scratch using only the remain set $R$, providing an ideal upper bound on $R$ accuracy and lower bound on $F$ accuracy, albeit at high computational cost. *Boundary Unlearning* implementation provided by authors of Chen et al. (2023) is equivalent to our method with $\lambda = 0$ and $\gamma = 0$ as in Eq. (3), where each forget sample is nudged across the decision boundary with a one-step gradient update, and then fine-tuned on the reassigned label. *Fine-tune on Remain* updates the original model using only data from $R$, leveraging distributional shift to induce forgetting but without direct boundary manipulation. *Negative Gradient* Golatkar et al. (2020) reverses learning on the forget set by optimizing the negative of the standard loss. *Catastrophic Forgetting-k* (CFK) and *Exact Unlearning-k* (EUK) Goel et al. (2022) freeze the first $k$ layers of the original model and fine-tune or retrain the remaining layers on $R$, offering controlled partial forgetting. Finally, *Scrub* Kurmanji et al. (2024) uses a student–teacher framework in which the student learns from a teacher trained solely on $R$, aiming to exclude information from $F$. All baseline comparisons are included in the main text results (Section 4).

# 4 Results

We have evaluated our unlearning method across benchmark, medical, and clinical imaging datasets to assess its ability to selectively forget designated data while preserving performance on the retained set. We have organized our results into four parts. First, we present quantitative results on four image classification datasets

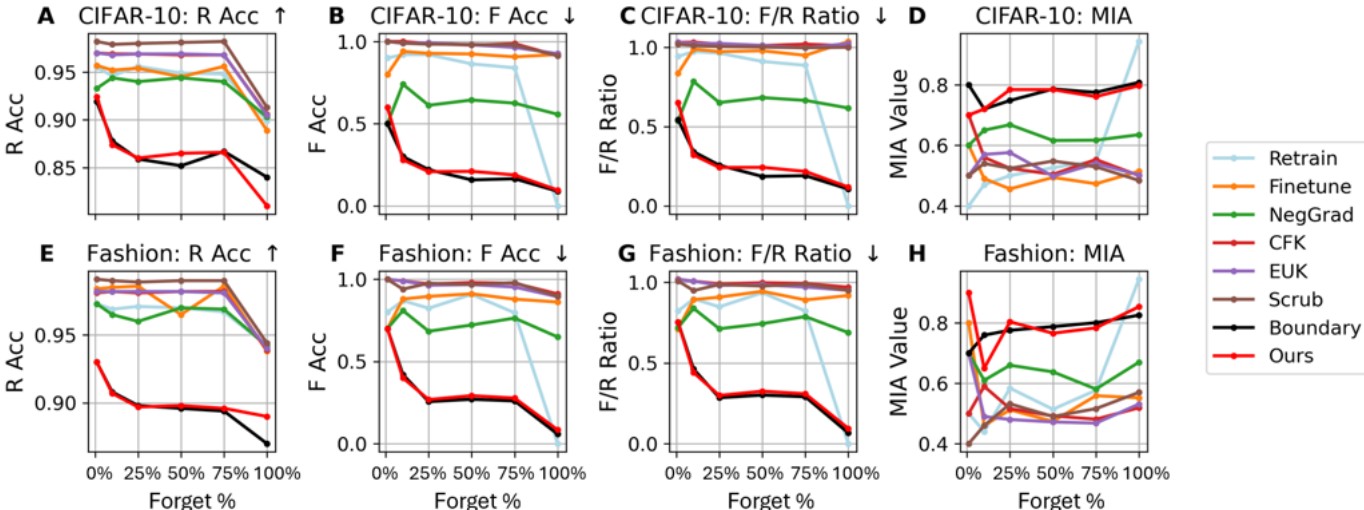

Figure 2: Unlearning varying proportions of a target class using our method and standard baselines. R Acc and F Acc denote the accuracy of the unlearned model on the remain and forget sets, respectively. MIA refers to robustness against membership inference attacks. Top row: CIFAR-10; bottom row: FashionMNIST. Arrows indicate the desirable direction for each metric (↑ for higher is better, ↓ for lower is better), and $MIA = .5$ indicates the best case.

(CIFAR-10, FashionMNIST, CFP-OS, and MRI), where we systematically unlearn varying proportions of the training data and evaluated unlearning success using four metrics: Forget Accuracy (F-Acc), Remain Accuracy (R-Acc), F/R ratio, and Membership Inference Attack (MIA) accuracy. Second, we examined real-world clinical scenarios, including attribute-based unlearning on color fundus photographs and external eye images. Third, we conducted ablation experiments to isolate the effects of key loss components and design choices within our bilevel optimization framework. For all datasets, we conducted a grid search over $\lambda \in [0, 1e-4, 1e-3, 1e-2, 1e-1]$ and $\gamma \in [0, 1e-4, 1e-1, 1]$ (as in Eq. (3)) and the best pairing were used. The best performing pair of $\gamma$ and $\lambda$ were used for reporting. Heatmaps of all pairings can be found in the Appendix (Figures 9,10,11).

## 4.1 Selective Forgetting

We first evaluate the ability of our algorithm to selectively forget varying proportions of a designated class in CIFAR-10 and FashionMNIST. For each dataset, we constructed forget sets $F$ by sampling increasing percentages of a single class (1%, 10%, 25%, 50%, 75% and 100%). We report Forget Accuracy (F-Acc), Remain Accuracy (R-Acc), F/R ratio, and Membership Inference Attack (MIA) accuracy in Figure 2.

Our method consistently achieves lower F-Acc than all baselines—often outperforming Chen et al. (2023), which is conceptually most similar. Across both datasets, this enhanced forgetting is achieved with only a modest reduction in R-Acc (typically 2–5%) compared to Fine-Tuning and CFK, which prioritize utility over explicit forgetting. Our formulation maintains a favorable tradeoff between forgetting and retention, as captured by a reduced F/R ratio. That said, we note that this tradeoff entails a measurable reduction in R-Acc compared to some other baselines.

As expected, when the forget set comprises only 1% of the data, full retraining is inefficient. While our method and Boundary Unlearning exhibit higher MIA values in these small-forget settings, they achieve markedly superior F-Acc suppression. On CFP-OS, we observe a small but consistent increase in R-Acc as the forget percentage approaches 100% (Appendix Figure 5). We believe that this is due to the effective reduction in class complexity, as the removal of nearly all examples from one class renders the task closer to binary classification. In such cases, the model no longer needs to learn fine-grained boundaries between three classes, allowing it to more confidently separate the remaining two, thereby improving R-Acc.

| Dataset | CFP-Clinic | | | | Oculoplastics | | | |
|---------|---------|---------|---------|---------|---------|---------|---------|---------|
| Unlearned | Camera | | | | VPF>12mm | | | |
| Metric | R Acc | F Acc | F/R | MIA | R Acc | F Acc | F/R | MIA |
| **Retrain** | **0.71** | 0.55 | 0.78 | $0.62 \pm 0.07$ | **0.97** | 1.00 | 1.03 | $0.50 \pm 0.00$ |
| **Finetune** | 0.65 | 0.81 | 1.25 | $0.59 \pm 0.06$ | 0.88 | 0.63 | 0.71 | $0.80 \pm 0.24$ |
| **NegGrad** | 0.64 | 0.86 | 1.35 | $0.55 \pm 0.04$ | 0.88 | **0.38** | 0.43 | $0.30 \pm 0.24$ |
| **CFK** | 0.70 | 0.79 | 1.12 | $0.57 \pm 0.06$ | 0.92 | 0.88 | 0.95 | $0.60 \pm 0.20$ |
| **EUK** | 0.70 | 0.79 | 1.13 | $\mathbf{0.50 \pm 0.09}$ | 0.94 | 0.88 | 0.93 | $\mathbf{0.50 \pm 0.00}$ |
| **Scrub** | 0.70 | 0.82 | 1.18 | $0.62 \pm 0.15$ | 0.91 | **0.38** | **0.41** | $0.50 \pm 0.32$ |
| **Boundary** | 0.49 | **0.09** | **0.19** | $0.74 \pm 0.05$ | 0.91 | 0.50 | 0.55 | $0.80 \pm 0.24$ |
| **Ours** | 0.51 | 0.10 | 0.20 | $0.58 \pm 0.10$ | 0.91 | **0.38** | **0.41** | $0.70 \pm 0.24$ |

Table 1: Performance of unlearning algorithms on clinical datasets. "Camera" (CFP-Clinic) refers to forgetting images captured with a specific imaging device, while $VPF > 12$" (Oculoplastic) denotes unlearning based on a clinical threshold for vertical palpebral fissure. R Acc = Remain set accuracy (higher is better), F Acc = Forget set accuracy (lower is better), F/R = Forget-to-Remain ratio (lower is better), and MIA = Membership Inference Attack accuracy (mean $\pm$ std). Bolded values indicate best performance column-wise (for MIA, values closest to .5)

.

When the forget set includes 100% of a class, the *Retrain* baseline achieves perfect forgetting by construction, as the class is entirely excluded from training. However, this scenario also results in high susceptibility to MIA, since the absence of this class makes its easier to detect by the attacker. Results from MRI and CFP-OS (Appendix Figure 5) mirror the trends observed in CIFAR-10 and FashionMNIST, confirming the consistency of our approach across domains. Additional numerical results are presented in Appendix Tables 5–10.

**Time Analysis**  On CIFAR-10, full retraining takes 1501.5 seconds. Unlearning with our method takes 134.2 seconds—an 11× speedup.

Additional timing results for medical datasets are provided in Appendix Figure 8.

## 4.2  Clinical Unlearning

We next evaluate our method in real-world clinical settings using two datasets: CFP-Clinic, comprising color fundus photographs labeled by disease class, and Oculoplastic, containing periocular images likewise labeled by disease class.

In contrast to Section 4.1, which focused on forgetting specific training samples from a given class, these experiments assess whether the model can forget the influence of clinically meaningful *subgroups* defined by acquisition device or anatomical features. Here we would like to preserve performance on similar but distinct examples. This setting mirrors realistic post-deployment revision needs in clinical AI systems. Metadata distributions for the targeted subgroups are shown in Appendix Figure 4. Importantly, while test samples were disjoint from training data, they shared the targeted properties. As such, in these experiments the Forget set ($F$) therefore refers to the held-out test samples that exhibit the same defining characteristic but were never part of the training data. The Remain set ($R$) corresponds to all other test samples that do not share this property. Thus, these experiments measure distributional unlearning i.e., the ability to remove learned associations tied to a subgroup rather than specific instances removal. Exact sample forgetting is addressed separately in our selective unlearning experiments.

We first evaluated unlearning based on acquisition device, removing all training images captured with a specific scanner (Cirrus 800 FA), a common scenario when a device is deprecated or recalled. Our method and Boundary Unlearning were the only approaches to reduce F-Acc to $\leq 10\%$, while maintaining comparable R-Acc (0.51 vs. 0.49). Although CFK preserved higher utility (R-Acc = 0.70), it failed to suppress F-Acc effectively, highlighting a utility–forgetting tradeoff that our method manages more explicitly. Overall R-Acc in this setting was lower than in other experiments (Table 1).

We also evaluated forgetting based on an anatomical criterion: vertical palpebral fissure (VPF) $\geq 12$ mm Dollfus & Verloes (2012). Increased VPF is a common feature of diseases such as Thyroid Eye Disease. Intuitively, this form of unlearning reflects evolving clinical criteria or quality thresholds Guimarães & Cruz (1995). Our approach achieved the best tradeoff in this setting, with low F-Acc (0.38) albeit slightly higher than retraining. However, our approach outperforms or is comparable to all other baselines considered. To our knowledge, this is the first demonstration of effective unlearning based on a continuous anatomical feature (Table 1).

Across both scenarios, our method consistently achieved the best or second-best F/R ratio, indicating targeted forgetting with minimal degradation to remain-set performance—surpassing conventional baselines and outperforming retraining. While privacy was not the primary goal in these settings, our method also achieved competitive membership inference (MIA) robustness. These results show that the model, after unlearning, generalizes the suppression of the targeted signal to unseen samples sharing that attribute, supporting its use as a practical maintenance mechanism on real world data.

### 4.2.1 Qualitative Comparison

To contextualize unlearning behavior, we present qualitative examples from each clinical unlearning scenario in Appendix Figure 6. For both the camera model and anatomical feature (VPF $\geq 12$ mm) experiments, we randomly selected pairs of test images from the same disease class: one from the forget set ($F$) and one from the remain set ($R$).

Qualitative results illustrate that the model was able to suppress correct classification on the targeted samples in $F$, while maintaining performance on visually similar examples in $R$. This suggests that unlearning operated not just at the class level, but in a clinically specific and metadata-informed way. In the camera-based unlearning example, images of the same disease captured with a different imaging device were retained. In the anatomical unlearning case, the model selectively suppressed predictions for high-VPF cases but preserved others within the same diagnostic category. Together, these examples demonstrate that the forgetting effect is both targeted and meaningful—removing learned associations tied to acquisition conditions or clinical phenotypes, rather than globally degrading the model.

### 4.2.2 Decision Boundary Shifts

To visualize how unlearning alters model behavior in clinical settings, we projected the embeddings of test samples in $F$ and $R$ using t-SNE and plotted approximate decision boundaries for the original, retrained, and unlearned models (Figure 3). Since t-SNE is initialized randomly, the absolute positions of class clusters may vary between runs. In the Oculoplastic dataset, unlearning images with vertical palpebral fissure (VPF) $\geq 12$ mm caused a visible shift in the decision space: points in the forget set that were originally classified as Class 2 (Thyroid Eye Disease) were reassigned to the nearest incorrect class (Class 0) in the unlearned model, consistent with our relabeling mechanism. Notably, this shift occurred without substantial distortion of the remain-set regions. A small number of points near the original decision boundary became misclassified as a side effect of the forgetting process, consistent with the expected tradeoff between retention and removal.

We also generated a corresponding decision boundary visualization for the CFP-Clinic dataset (Appendix Figure 7). Unlike the Oculoplastic setting, the baseline model for CFP exhibited poor separability and high prediction uncertainty across all classes. So, the t-SNE embeddings produced diffused and overlapping decision regions. Further discussion on this can be found in Appendix Section A.7.2. In summary, quantitative metrics showed a consistent drop in F-Acc following unlearning, and qualitative examples confirmed that targeted images were misclassified as intended (Table 1).

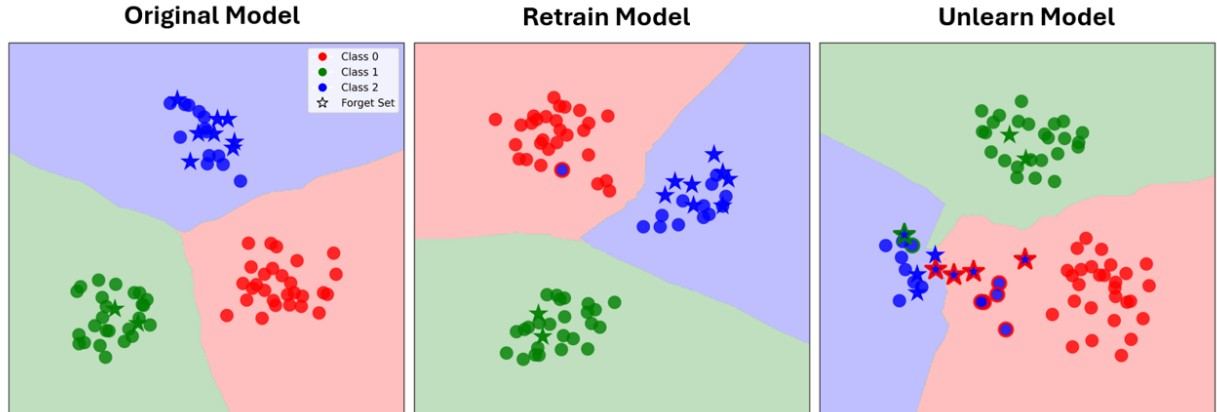

Figure 3: Decision boundary of original (left), retrain (center), and model unlearned of images with VPF $\geq$ 12 mm (right). All points in the test set for $R$ and $F$ were passed through the models and the embeddings were visualized using t-SNE. Stars denote points in $F$, and circles denote points in $R$. Color denotes predicted label, and in the event of misclassification, edges denote the ground truth label. Decision space was visualized by training a $k$-nearest neighbors classifier on the 2D t-SNE embeddings using predicted labels, and plotting its decision regions as background contours.

**Retraining and unlearning.** Retraining from scratch is widely regarded as the gold standard for unlearning, as it guarantees that the model has no direct exposure to the forget set. However, in our experiments (Figure 3, Table 1), we observed that retraining did not always lead to low accuracy on $F$. This is expected when $F$ contains samples that are highly representative of their class, since the model can still rely on similar examples present in the retain set. Our method differs in that it provides an explicit mechanism to suppress predictive influence from $F$, offering direct control over the unlearned model.

## 4.3 Ablation Studies

To assess the impact of key design choices in our outer optimization, we conducted ablation experiments on CIFAR-10 using a fixed forget set (100% of class 0). We evaluated three factors: (1) whether losses were computed using logits or standard hard labels (argmax), (2) whether logits were preprocessed via top-$k$ normalization, and (3) whether loss from the remain set $R$ was incorporated during unlearning, either from the beginning or with a delayed onset. Details on the settings modified in ablation experiments can be found in Section 2.4

Remain loss was weighted using a regularization parameter $\phi$ as in Eq. (1), set to $\phi = 10^{-2}$ and $\lambda = \in [0, 10^{-1}, 10^{-4}]$ was evaluated as in Eq. (3). $\gamma$ was fixed at 0 for all ablation studies. We also varied how logits were used on both $F$ and $R$ as described in Eq. (17). Results are summarized in Table 2, with additional configurations presented in Appendix Table 4.

Computing loss on $F$ using unprocessed logits leads to performance degradation on $R$ (R-Acc = 0.17), despite strong forgetting. Including remain loss generally improves $R$ accuracy across configurations, at the expected cost of reduced forgetting. Delaying the introduction of remain loss (e.g., starting at epoch 10) showed negligible difference compared to early inclusion. Notably, our default setting i.e., argmax labels for $F$, no logits, and no remain loss, achieves the best combination of forgetting and retention (F-Acc = 0.094, R-Acc = 0.81).

These results underscore the method's sensitivity to outer loop settings. Notably, seemingly small adjustments can produce large performance shifts. However, this allows the model to be steered toward different points along the forgetting–retention spectrum in a predictable way. In particular, using logits from the forget set enforces strong forgetting at the expense of remain accuracy, while incorporating logits from the remain set boosts retention but weakens forgetting. Preprocessing logits provides a minor stabilizing effect. Together,

these findings show that our method offers tunable (but not always stable) control over the forgetting–retention spectrum.

| R Logit | F Logit | Preproc. | F Acc | | | R Acc | | |
|---|---|---|---|---|---|---|---|---|
| | | | $\lambda = 0$ | $\lambda = 1e-4$ | $\lambda = 1e-1$ | $\lambda = 0$ | $\lambda = 1e-4$ | $\lambda = 1e-1$ |
| − | − | − | 0.089 | 0.12 | 0.094 | 0.84 | 0.81 | 0.81 |
| + | − | − | 0.49 | 0.52 | 0.48 | 0.87 | 0.85 | 0.85 |
| − | + | − | 0.001 | 0.047 | 0.062 | 0.18 | 0.17 | 0.17 |
| + | + | − | 0.001 | 0.047 | 0.062 | 0.18 | 0.17 | 0.17 |
| +(80) | +(80) | − | 0.001 | 0.047 | 0.062 | 0.18 | 0.17 | 0.17 |
| +(90) | +(90) | − | 0.001 | 0.047 | 0.062 | 0.18 | 0.17 | 0.17 |
| + | + | + | 0.52 | 0.55 | 0.51 | 0.89 | 0.88 | 0.89 |

Table 2: Evaluation of the influence of logits and incorporation of the loss from $R$ in unlearning. A $\phi$ value of $1e-2$ was used. Results with $\phi = 1e-4$ are in Appendix Table 4. A number after the plus sign in parentheses denotes the epoch at which the logits were incorporated during outer optimization.

## 5 Discussion

Machine unlearning is often framed as a privacy-preserving tool to satisfy regulatory mandates like the "right to be forgotten" Regulation (2020); Graves et al. (2021). In clinical ML systems, where data heterogeneity, evolving standards, and device turnover are common, full retraining is often infeasible—particularly for deep architectures Guo et al. (2021); Moreno-Torres et al. (2012); Futoma et al. (2020). To address this, we propose a bilevel optimization algorithm with convergence guarantees that enables selective removal of training data from convolutional neural networks. Our experiments span both benchmark and clinical datasets, each designed to evaluate *targeted unlearning* under different operational constraints. Our settings reflect realistic maintenance scenarios and test whether a model can forget a subgroup's influence rather than specific samples. In all cases, our approach suppressed classification performance on the targeted group while preserving retention performance, demonstrating reliable behavioral control.

The motivation for our clinical unlearning experiments is not to claim that data from specific devices or patients with specific features is intrinsically detrimental, but rather to illustrate how unlearning can be applied in clinically realistic maintenance scenarios. In practice, medical imaging systems frequently face distribution shifts when equipment is upgraded or replaced, and such shifts have been shown to degrade model performance if not addressed Guan & Liu (2021); Kumari & Singh (2024). Our experiment therefore serves as a proof-of-principle demonstration that targeted device-level unlearning is feasible. While we do not empirically show that retaining this particular device's data harms future performance, our results demonstrate the mechanism by which device-specific removal can be enacted when clinically motivated. Future work could more directly quantify the performance tradeoffs associated with device retention versus removal.

Although privacy is not the primary motivation of our framework, we evaluated robustness to MIA for completeness. We observed that our method sometimes yields higher MIA scores than retraining or other baselines. One possible explanation is that explicit boundary-shifting may leave sharper or less smooth decision regions around forget samples, which could be exploited by adversaries in membership inference. To our knowledge, prior boundary-based methods did not report MIA experiments, so further work is needed to fully characterize this tradeoff. These results underscore that while our method enables targeted forgetting, it is not designed as a privacy-preserving unlearning algorithm. Future extensions could integrate MIA robustness more directly for settings where privacy, rather than model maintenance, is the primary concern.

While our framework offers tunable behavioral control, the ablation study highlights that this tunability comes with sensitivity to hyperparameters. In practice, we found that the effects of different settings follow interpretable trends: (i) using argmax labels for forget samples with no remain loss maximizes forgetting but sacrifices retention; (ii) incorporating remain loss with moderate weight improves retention while reducing forgetting. These heuristics allow practitioners to steer the method toward desired outcomes without exhaustive search. Nonetheless, tuning can remain challenging in high stakes or resource constrained settings, as small parameter changes may still cause large shifts. Developing more robust strategies for parameter selection is therefore an important direction for future work.

We have identified some limitations of our approach. First, our method is tailored to CNNs, and adaptation to other architectures such as vision transformers (ViTs) remains an open challenge due to their use of attention and context-rich feature representations Shao et al. (2021); Mahmood et al. (2021). Second, in distributionally entangled datasets, such as when unlearning glaucoma cases tied to a deprecated scanner, forgetting can degrade performance on the remain set (Appendix Figure 4). This highlights challenges in separating acquisition effects from disease labels. In the future, it would be interesting to see if our bilevel formulation can be adapted to unlearn vision transformers (ViTs). The main challenge here is that ViTs rely on attention mechanisms and context-aware representations less amenable to boundary perturbations Shao et al. (2021); Mahmood et al. (2021) that need to be accounted in the formulation. Furthermore, additional experiments on larger-scale clinical models are warranted, especially across diverse imaging modalities (e.g., CT, OCT, histopathology) and deployment conditions (e.g., continual or federated learning). Finally, we note that our definition of forgetting (requiring misclassification of $F$) differs from the retraining-based standard of statistical indistinguishability. While this provides a practical guarantee for model maintenance, it may be less appropriate in privacy-driven settings.

## 6 Implications of Unlearning in Healthcare

Our formulation of unlearning as enforced misclassification raises important ethical considerations in high-stakes domains such as healthcare. If a patient invokes their right to be forgotten, an unlearned model could actively misclassify future data from that patient. Similarly, when unlearning is applied for maintenance purposes (e.g., removing data from deprecated imaging devices or outlier subgroups), there is a risk of inadvertently propagating biases against those populations. We emphasize that our contribution is methodological and not intended as a guide book for immediate clinical deployment. Any adoption of unlearning in healthcare would require strict safeguards, including registries to track which patients or subgroups have been unlearned, governance mechanisms to ensure unlearned models are not inappropriately applied for diagnostic purposes, and oversight to mitigate bias introduction.

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

# A  Appendix

## A.1  Proof of Lemma 2.2

**Boundary point.** Conceptually, our approach to solving the inner optimization problem involves using the Lagrangian of the constrained problem and then solving the equivalent unconstrained form which is easier to implement in practice. Given $f(w_0, \delta + x_i) \in [0, 1]^K$ to be class probability predictions, and $y_i = e_k$ for some $k$ (i.e., $e_k$ is the $k-$th standard basis vector), the inner optimization is a optimization problem with a single inequality constraint given by,

$$\min_\delta 0 \ \ \text{subject to} \ \ f(w_0, \delta + x_i)^\top e_k \leq \kappa.$$

Recall that for classification tasks, we use $\mathcal{L} = -\log(\cdot)$ (i.e., cross entropy based loss) which is a monotonic decreasing function in the positive real numbers. Therefore, the above problem is equivalent to,

$$\min_\delta 0 \ \ \text{subject to} \ \ \mathcal{L}(f_k(w_0, \delta + x_i)) \geq \mathcal{L}(\kappa). \tag{5}$$

Now we write down the lagrangian $H$ of (5) as,

$$L(\delta, u) = u \cdot (\mathcal{L}(\kappa) - \mathcal{L}(f_k(w_0, \delta + x_i))), \tag{6}$$

where $u \geq 0$ is the lagrange multiplier with dual feasibility constraint. We can ignore the second term of $\kappa$ since it is not a function of $\delta$. Moreover, since it is sufficient to find an approximately optimal $\delta$, we can assume that $u > 0$, that is, the constraint (5) is active.

This lagrangian $L$ has to be minimized wrt $\delta$ and maximized wrt to $u$. Using the fact that $\inf(-F) = \sup F$ for any function $F$, we have the desired result. Aside, the complementarity slackness condition necessitates that $u \cdot (\mathcal{L}(\kappa) - \mathcal{L}(f_k(w_0, \delta^* + x_i))) = 0$ so if $u > 0$, then the inequality constraint is tight - and we so we obtain a "boundary" point.

For regression tasks with squared $\ell_2$ norm loss, $y_i \in \mathbb{R}^K$ we may modify the inner optimization as,

$$\min_\delta 0 \ \ \text{subject to} \ \ \|f(w_0, \delta + x_i) - y_i\|_2^2 \geq \kappa,$$

and proceed with the lagrangian as above.

**Closest boundary point.** We can show that this approach works for any inner objective $l(\delta)$ that is smooth (or subdifferentiable) and $\lambda \geq 0$ (not necessary to have $\lambda = 0$). To this end, consider the case when $l(\delta) = \frac{1}{2}\|\delta\|_2^2$ that was used in Chen et al. (2023). Our proof of Lemma 3.2 can easily handle this. The Lagrangian to be minimized wrt $\delta$ in this case becomes,

$$L(\delta, u) = \frac{1}{2}\|\delta\|_2^2 + u \cdot (\mathcal{L}(\kappa) - \mathcal{L}(f_k(\delta + x_i, w_0))),$$

and the gradient wrt $\delta$ is given by $\delta - u \cdot \nabla_\delta \mathcal{L}(f_k(\delta + x_i, w_0))$. Note that since there is only one constraint, the dual variable $u \geq 0$ is a scalar which we consider as a hyperparameter. With this, we can scale the objective by $u$, and minimize

$$\mathcal{L}(\delta) = \frac{1}{2u}\|\delta\|_2^2 - \mathcal{L}(f_k(\delta + x_i, w_0)),$$

or maximize

$$-\mathcal{L}(\delta) = -\frac{1}{2u}\|\delta\|_2^2 + \mathcal{L}(f_k(\delta + x_i, w_0)).$$

Note that the gradient of $\mathcal{L}$ wrt $\delta$ is given by $\frac{1}{2u}\delta - g$.

Convergence for $\lambda \neq 0$: Notation-wise in (3), we use $\lambda = \frac{1}{u} > 0$ which we consider as a hyperparameter. So for any fixed $\lambda \geq 0$, the inner optimization problem is simply given by,

$$\max_{\delta} -\frac{\lambda}{2}\|\delta\|_2^2 + \mathcal{L}(f_k(\delta + x_i, w_0)).$$

Whence, the convergence result in Safaryan & Richtárik (2021) holds as we can get the exact gradient of the term $\lambda\|\delta\|_2^2$. So as long as the objective function is a deterministic function of the perturbation $\delta$, our convergence analysis is valid.

## A.2 Proof of Corollary 2.3

We will suppress some notation to make the exposition simpler. For example, we will use $\nabla_\delta \mathcal{L}$ to denote $\nabla_\delta \mathcal{L}(f_k(w_0, \delta + x_i))$, and $\delta^t$ to denote $t-$th iterate of $\delta$ in inner loop. In this notation, update rule for Boundary Shrink in Chen et al. (2023) is given by,

$$x^{t+1} = x^t + \epsilon \operatorname{sign}\left(\nabla_x \mathcal{L}(x^t)\right), x^0 = x_i, i \in F. \tag{7}$$

On the other hand, sign gradient ascent with step size/learning rate $\epsilon$ on $\mathcal{L}(x_i + \delta)$ (with no closeness regularization) is given by,

$$\delta^{t+1} = \delta^t + \epsilon \operatorname{sign}\left(\nabla_\delta \mathcal{L}(x_i + \delta^t)\right), \delta^0 = 0, i \in F. \tag{8}$$

Note that if we set $x = \delta + x_i$ (in (8)), then $x^t = \delta^t + x_i$, and the jacobian of $x$ wrt $\delta$ is identity matrix. Moreover, using chain rule, we have that,

$$\nabla_\delta \mathcal{L}(x_i + \delta^t) = \nabla_\delta x \cdot \nabla_x \mathcal{L}(x^t) = \nabla_x \mathcal{L}(x^t). \tag{9}$$

Adding $x_i$ to both the sides of our update rule (8), we have the desired result.

## A.3 Proof of Lemma 2.4

We first compute $\mathbb{E}[d]$ and so assume that $\epsilon = 1$ – this is without of loss of generality since $\mathbb{E}$ is a linear operator. Since $z$ has a continuous density function, $\mathbb{P}(z = -g/\gamma) = 0$, and since the sign is computed elementwise/coordinatewise, we will compute $d$ coordinatewise. So we have that,

$$\mathbb{E}_z[d] = \mathbb{E}[\operatorname{sign}(g + \gamma z)] = +1 \cdot \mathbb{P}\left(z \geq -\frac{g}{\gamma}\right) - 1 \cdot \mathbb{P}\left(z \leq -\frac{g}{\gamma}\right)$$

$$= 1 - 2 \cdot \mathbb{P}\left(z \leq -\frac{g}{\gamma}\right) = 1 - 2\left[\frac{1}{2}\left(1 + \operatorname{erf}\left(\frac{-g}{\gamma\sqrt{2}}\right)\right)\right] = \operatorname{erf}\left(\frac{g}{\gamma\sqrt{2}}\right), \tag{10}$$

where we used the definition of CDF of $z$ using $\operatorname{erf}(\cdot) \in (-1, +1)$, the Gaussian Error Function. Note that $\operatorname{erf}(g) > 0$ if and only if $g > 0$, that is, $g$ and $\operatorname{erf}(g)$ share the same sign along all the coordinates. So, we have that $\mathbb{E}_z[d^\top g] = \mathbb{E}_z[d]^\top g \geq 0$ finishing the proof. By substituting the CDF of Laplacian or Cauchy in equation (10), we can see that the results holds for these distributions also.

## A.4 Proof of Theorem 2.5

First, we will use $l$ to denote $-L$. So the inner optimization becomes the following minimization problem,

$$\max_{\delta} L(\delta) = \min_{\delta} -\mathcal{L}(\delta) = \min_{\delta} l(\delta). \tag{11}$$

Since $l$ is differentiable and satisfies lipschitz assumption, we have that, for any $\lambda, \delta \in \mathbb{R}^D$,

$$l(\lambda) \leq l(\delta) + \nabla l(\delta)^\top (\lambda - \delta) + \frac{L}{2}\|\lambda - \delta\|_2^2. \tag{12}$$

By setting $\lambda = \delta^{t+1}$, and $\delta = \delta^t$ in the above inequality, we get

$$l(\delta^{t+1}) \leq l(\delta^t) - \epsilon_t \nabla l(\delta^t)^\top \text{sign}(\nabla_\delta l(\delta^t) + \gamma \cdot z) + \frac{L\epsilon_t^2}{2} \| \text{sign}(\nabla_\delta l(\delta^t) + \gamma \cdot z) \|_2^2$$

$$\leq l(\delta^t) - \epsilon_t \nabla l(\delta^t)^\top \text{sign}(\nabla_\delta l(\delta^t) + \gamma \cdot z) + \frac{LD\epsilon_t^2}{2}, \tag{13}$$

where the second inequality becomes an equality almost surely. Taking expectations with respect to $z$ on both sides of the inequality in (13) we get,

$$\mathbb{E}[l(\delta^{t+1})] \leq \mathbb{E}[l(\delta^t)] - \epsilon_t \nabla l(\delta^t)^\top \text{erf}\left(\frac{\nabla l(\delta^t)}{\gamma\sqrt{2}}\right) + \frac{LD\epsilon_t^2}{2}$$

$$= \mathbb{E}[l(\delta^t)] - \epsilon_t \left|\nabla l(\delta^t)\right|^\top \left|\text{erf}\left(\frac{\nabla l(\delta^t)}{\gamma\sqrt{2}}\right)\right| + \frac{LD\epsilon_t^2}{2}, \tag{14}$$

where we used the fact that $\nabla l$ and $\text{erf}(\nabla l)$ have the same sign, and that erf is an odd function. Subtracting $\inf_\delta l(\delta)$ from both the sides of the inequality, rearranging, and taking expectation with respect to all the perturbations added, we get,

$$\epsilon_t \mathbb{E}\left[\left|\nabla l(\delta^t)\right|^\top \text{erf}\left(\frac{|\nabla l(\delta^t)|}{\gamma\sqrt{2}}\right)\right] = \epsilon_t \left|\nabla l(\delta^t)\right|^\top \left|\text{erf}\left(\frac{\nabla l(\delta^t)}{\gamma\sqrt{2}}\right)\right|$$

$$\leq \mathbb{E}[l(\delta^t) - \inf l] - \left(\mathbb{E}[l(\delta^{t+1})] - \inf l\right) + \frac{LD\epsilon_t^2}{2}. \tag{15}$$

Summing up the inequalities in (15) from $t = 0$ to $t = T - 1$ we get,

$$\sum_{t=0}^{T-1} \epsilon_t \mathbb{E}\left[\left|\nabla l(\delta^t)\right|^\top \text{erf}\left(\frac{|\nabla l(\delta^t)|}{\gamma\sqrt{2}}\right)\right] \leq \mathbb{E}[l(\delta^0) - \inf l] - \left(\mathbb{E}[l(\delta^T)] - \inf l\right) + \sum_{t=0}^{T-1} \frac{LD\epsilon_t^2}{2}$$

$$= l(\delta^0) - \inf l - \left(\mathbb{E}[l(\delta^T)] - \inf l\right) + \frac{LD\epsilon_t^2}{2}$$

$$\leq l(\delta^0) - \inf l + \sum_{t=0}^{T-1} \frac{LD\epsilon_t^2}{2}, \tag{16}$$

where we used the fact $\delta_0$ is initialized at 0, and that $\left(\mathbb{E}[l(\delta^T)] - \inf l\right) \geq 0$. By choosing $\gamma$ to be the coordinatewise minimum value of gradient magnitude $|\nabla l(\delta^t)|$, we can assure that $|\nabla l(\delta^t)|^\top \text{erf}\left(\frac{|\nabla l(\delta^t)|}{\gamma\sqrt{2}}\right) = c\|\nabla l\|_1$, and the claim follows by following similar techniques as in Garrigos & Gower (2023) Theorem 5.12, since the step size/learning rate $\epsilon_t = 1/t$.

### A.5 New Heuristics for Improved Unlearning

Our formulation provides three practical extensions. First, the remain set loss can be incorporated during the outer loop to preserve performance on $R$:

$$w_u = \arg\min_w \sum_{i \in \tilde{F}} \mathcal{L}(f_w(x_i), y_i^b) + \phi \sum_{i \in R} \mathcal{L}(f_w(x_i), y_i). \tag{17}$$

The proportion of the loss from $R$ incorporated is controlled by $\phi$.

Second, we allow the outer loss to be computed using dense logits instead of hard labels. To interpolate between sparse (argmax) and dense supervision, we define a simple top-$k$ logit preprocessing step. In all experiments, $k$ was set to 3. In detail, let $f_w(x_i) \in \mathbb{R}^K$ denote the logits for sample $x_i$, and let Top-$k(f_w(x_i))$ denote the indices of the largest $k$ logits. We define a sparse logit vector $\tilde{f}_w(x_i)$ as:

$$\tilde{f}_{w,j}(x_i) = \begin{cases} f_{w,j}(x_i), & \text{if } j \in \text{Top-}k(f_w(x_i)) \\ 0, & \text{otherwise} \end{cases}$$

We then normalize the retained logits to form a soft target:

$$\tilde{p}_j(x_i) = \frac{\tilde{f}_{w,j}(x_i)}{\sum_{j' \in \text{Top-}k} \tilde{f}_{w,j'}(x_i)}.$$

These soft targets $\tilde{p}(x_i)$ are used in place of one-hot labels during fine-tuning. This modification reduces unnecessary updates to weights associated with classes far from the decision boundary. These settings—including top-$k$ logit selection, the use of soft versus hard labels, and whether the remain loss is included—are systematically explored in our ablation experiments (see Section 4.3).

Finally, by modifying the settings within our formulation, the unlearning process (as in Eqs. (3), eq:outopt) can be steered toward different performance tradeoffs. When unlearning is significantly more efficient than retraining from scratch, it becomes feasible to generate multiple unlearned models—each optimized for distinct objectives (e.g., high accuracy on $R$ at the expense of $F$, and vice versa). We have included the pseudocode of our outer loop with the above three extensions in the Appendix (Algorithm 3).

### A.6 On the threshold $\kappa = 1/K$

In Equation (1), we set the boundary threshold to $\kappa = 1/K$. This choice ensures that the true class $y$ is no longer strictly dominant:

1. If $p_y(x) > 1/K$, class $y$ may still be the maximizer.

2. If $p_y(x) = 1/K$, then by the pigeonhole principle at least one $j \neq y$ satisfies $p_j(x) \geq 1/K$, meaning the point lies on a boundary (via a tie) or is already overtaken.

3. If $p_y(x) < 1/K$, then $y$ is strictly dominated and the point is across the boundary.

Thus, $\kappa = 1/K$ guarantees that the perturbed point is no longer in the interior of class $y$'s region, but lies on or across a decision boundary. In the binary case ($K = 2$), this coincides exactly with the pairwise boundary $p_y = p_j$.

### A.7 Additional Results and Details on the Methods

In the following sections, we present additional experiments and implementation details that supplement the main results. These include dataset summaries, extended unlearning benchmarks, timing analysis, hyperparameter sweeps, and the raw data underlying selective unlearning plots.

#### A.7.1 Datasets

We used a diverse set of datasets spanning both open-source and clinical imaging domains. This section outlines the composition of the open source CFP dataset that we curated as well as our clinical datasets (Appendix Table 3. For clinical experiments, we selected data based on imaging modality and anatomical measurements, ensuring real-world relevance (Appendix Figure 4).

|  | IDRiD | RFMID | ORIGA | G1020 | Total |
|---|---|---|---|---|---|
| **DR** | 260 | 240 | 0 | 0 | 500 |
| **Glaucoma** | 0 | 0 | 168 | 296 | 464 |
| **Other** | 0 | 500 | 0 | 0 | 500 |

Table 3: Open source data details. For DR, 260 images from the Indian Diabetic Retinopathy Image Dataset Porwal et al. (2018) with severity of $> 2$ were chosen, and 240 images having a diagnosis of only DR from the Retinal Fundus Multi-Disease Image dataset (RFMID) Pachade et al. (2020) were selected. For Glaucoma, 168 images from the Online Retinal Fundus Image Database for Glaucoma Zhang et al. (2010) and 296 images from the G1020 Bajwa et al. (2020) dataset were selected. 'Other' images were selected by randomly sampling 500 images from RFMID that did not have DR or Glaucoma as a label. These images have diagnoses of AMD, macular hole, and more Bajwa et al. (2020).

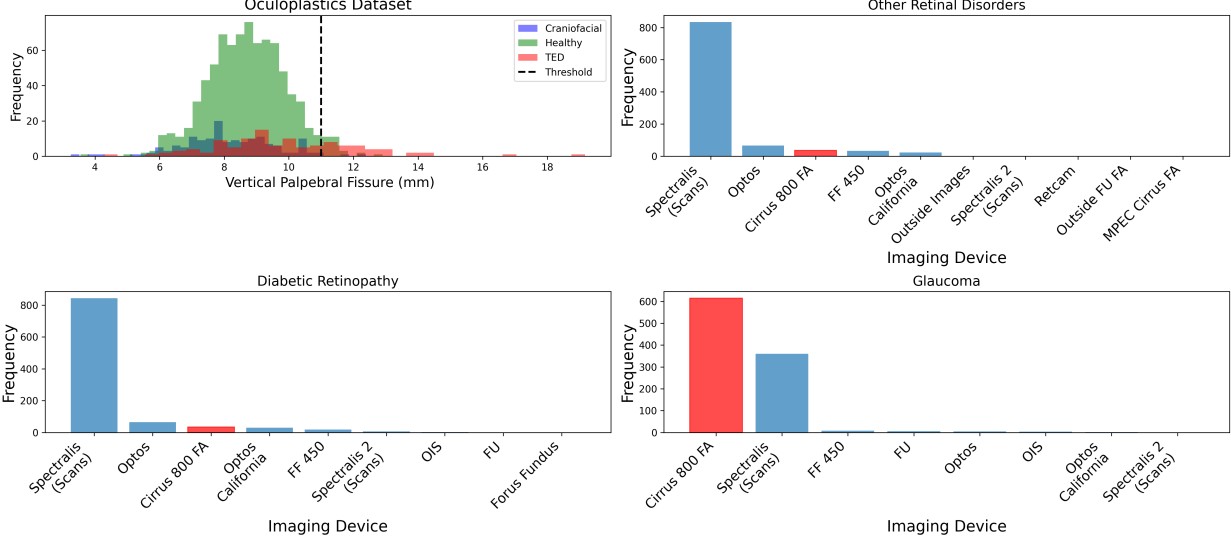

Figure 4: Distributions of clinical data and parameters for unlearning used in this study. On the histogram, the dashed line denotes vertical palpebral fissure (VPF) of 11- images with a VPF greater than this were unlearned. On bar graphs, 'red' denotes the samples that were unlearned in targeted medical unlearning experiments.

#### A.7.2 Additional Unlearning

In this section, we show extended results from class-wide unlearning across two domains: 2D brain MRI slices and fundus images (CFP). We report forgetting accuracy (F-Acc), retention accuracy (R-Acc), their ratio, and mean ± std of MIA attack success (Appendix Figure 5). We also provide a pseudocode algorithm of our

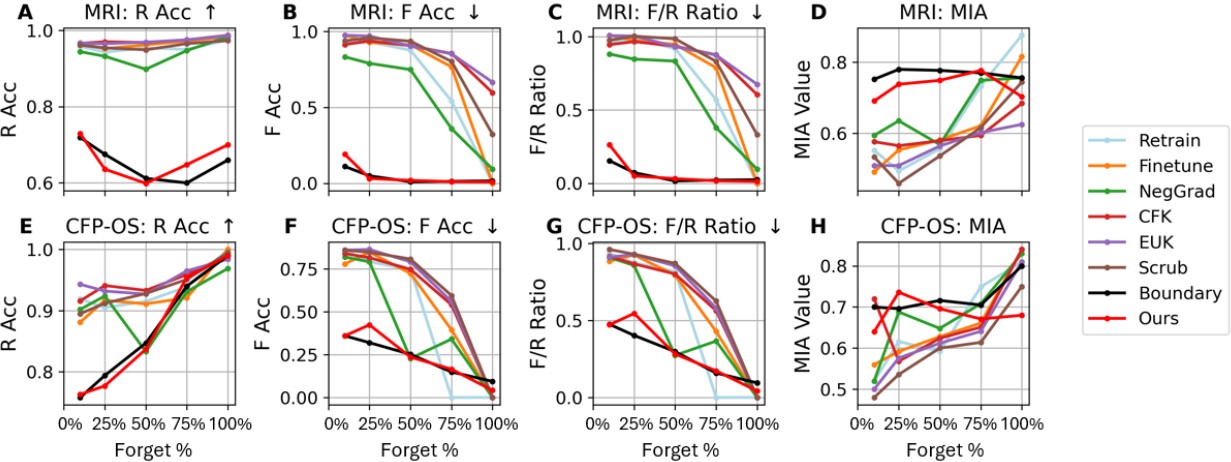

Figure 5: Unlearning various proportions of the class to be forgotten using our method and other standard baselines. R and F acc denote the accuracy of the unlearned model on the R and F subsets. MIA denotes robustness to membership inference attacks. Top row displays results on MRI dataset, bottom row displays results on CFP open source dataset. Directionality of arrows indicates whether high or low values are the desired output.

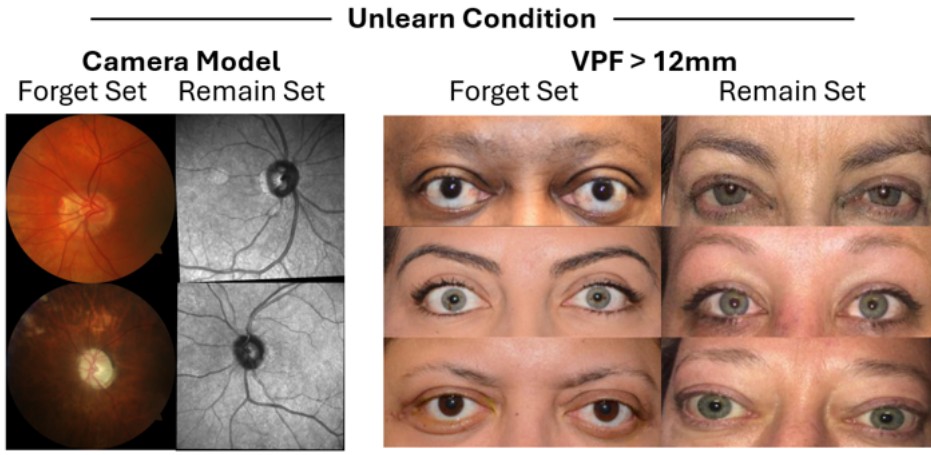

Figure 6: Example images from both clinical unlearning scenarios, drawn from the forget set ($F$) and remain set ($R$). All images belong to the same disease class, but forget samples were either collected using the Cirrus 800 FA imaging device (left) or had a vertical palpebral fissure (VPF) $\geq$ 12mm (right), and were explicitly targeted for unlearning.

outer loop of optimization with the extensions studied in ablation experiments. Full results on ablation using different $\phi$ and $\lambda$ values are reported in Tables 2 and 4.

For completeness, we include a t-SNE visualization of the decision boundary shift following unlearning on the CFP-Clinic dataset (Figure 7). In contrast to the Oculoplastic dataset, the CFP-Clinic embeddings exhibited poor baseline separability and high uncertainty in predictions across all models. As a result, the decision regions appear diffuse, and no clearly structured shift is visible after unlearning. This reflects the low-quality latent space induced by the initial model, rather than a failure of the unlearning algorithm itself. In fact, forgetting metrics still showed a consistent drop in F-Acc (Table 1), and qualitative examples confirm that forgotten images were suppressed even when embedded in noisy regions. This example highlights both the utility and limitations of decision-boundary visualization in low-performing clinical domains.

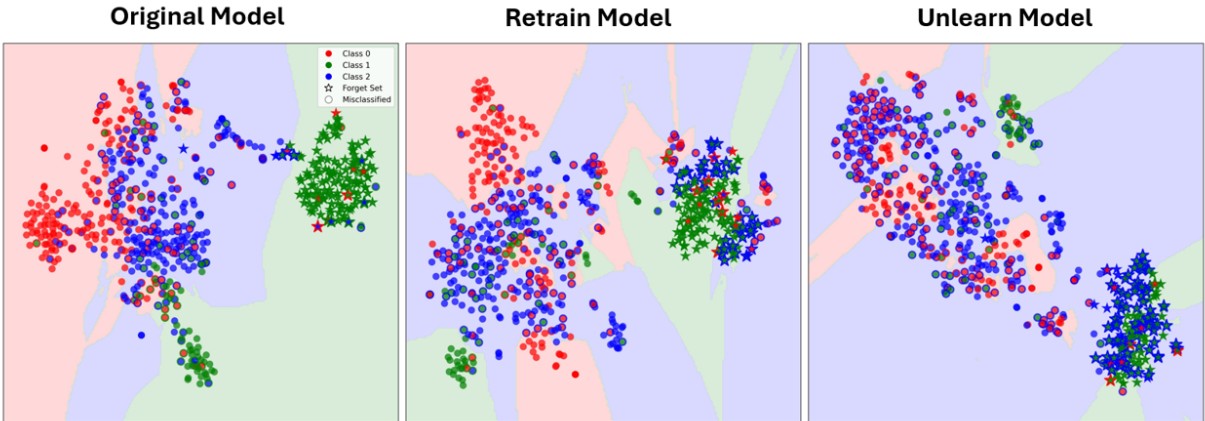

Figure 7: Decision boundary of original (left), retrain (center), and unlearned (right) models. All points in the test set for $R$ and $F$ were passed through the models and the embeddings were visualized using t-SNE. Stars denote points in $F$, and circles denote points in $R$. Color denotes predicted label, and in the event of misclassification, edges denote the ground truth label. Decision space was visualized by training a $k$-nearest neighbors classifier on the 2D t-SNE embeddings using predicted labels, and plotting its decision regions as background contours.

Table 4: Evaluation of the influence of logits and incorporation of the loss from $R$ in unlearning. A $\phi$ value of $1e-4$ was used. A number after the plus sign in parentheses denotes the epoch at which the logits were incorporated during outer optimization.

| | | | F Acc | | | R Acc | | |
|---|---|---|---|---|---|---|---|---|
| R Logit | F Logit | Preproc. | $\lambda = 0$ | $\lambda = 1e-4$ | $\lambda = 1e-1$ | $\lambda = 0$ | $\lambda = 1e-4$ | $\lambda = 1e-1$ |
| − | − | − | 0.089 | 0.12 | 0.094 | 0.84 | 0.81 | 0.81 |
| + | − | − | 0.49 | 0.52 | 0.48 | 0.87 | 0.85 | 0.86 |
| − | + | − | 0.001 | 0.047 | 0.062 | 0.18 | 0.17 | 0.17 |
| + | + | − | 0.001 | 0.047 | 0.062 | 0.18 | 0.17 | 0.17 |
| +(80) | +(80) | − | 0.001 | 0.047 | 0.062 | 0.18 | 0.17 | 0.17 |
| +(90) | +(90) | − | 0.001 | 0.047 | 0.062 | 0.18 | 0.17 | 0.17 |
| + | + | + | 0.44 | 0.49 | 0.49 | 0.88 | 0.87 | 0.88 |

---

**Algorithm 3:** Configurable Outer Loop with Remain Loss and Logit-Based Labeling

---

**Input:** Relabeled forget set $\tilde{F}$, Remain set $R$, initial weights $w_0$, learning rate $\eta$,
    remain loss weight $\phi$, top-$k$ parameter $k$, `use_soft_labels`, `use_remain_loss`
**Output:** Unlearned model parameters $w_u$
Initialize $w \leftarrow w_0$;
**for** $t = 1$ **to** $T_{outer}$ **do**
    Sample minibatch $(x_f, y_f^b) \sim \tilde{F}$;
    ;                                     // Load forget-set samples
    **if** *use_soft_labels* **then**
        $z_f \leftarrow f_{w_0}(x_f^b)$ ;                // Compute logits from original model
        Keep top-$k$ entries of $z_f$, zero others;
        Normalize retained logits to form $\tilde{p}_f$;
        ;                    // Create soft target distribution
        $\mathcal{L}_f \leftarrow \mathcal{L}(f_w(x_f), \tilde{p}_f)$ ;     // Compute outer loss with soft targets
    **else**
        $\mathcal{L}_f \leftarrow \mathcal{L}(f_w(x_f), y_f^b)$ ;             // Use hard labels
    **if** *use_remain_loss* **then**
        Sample minibatch $(x_r, y_r) \sim R$;
        ;                    // Load remain-set samples
        $\mathcal{L}_r \leftarrow \mathcal{L}(f_w(x_r), y_r)$;
        ;                    // Compute loss on retained data
        $\mathcal{L}_{\text{total}} \leftarrow \mathcal{L}_f + \phi \cdot \mathcal{L}_r$;
        ;                    // Weighted combination
    **else**
        $\mathcal{L}_{\text{total}} \leftarrow \mathcal{L}_f$;
    Update $w \leftarrow w - \eta \cdot \nabla_w \mathcal{L}_{\text{total}}$;
    ;                                     // Perform gradient step
**return** $w_u \leftarrow w$

---

### A.7.3   Timing

We report wall-clock timing of unlearning procedures as a function of forget set size for each clinical dataset. As shown in Appendix Figure 8, our method scales efficiently even at high forget ratios, with runtimes remaining significantly lower than full retraining in all cases.

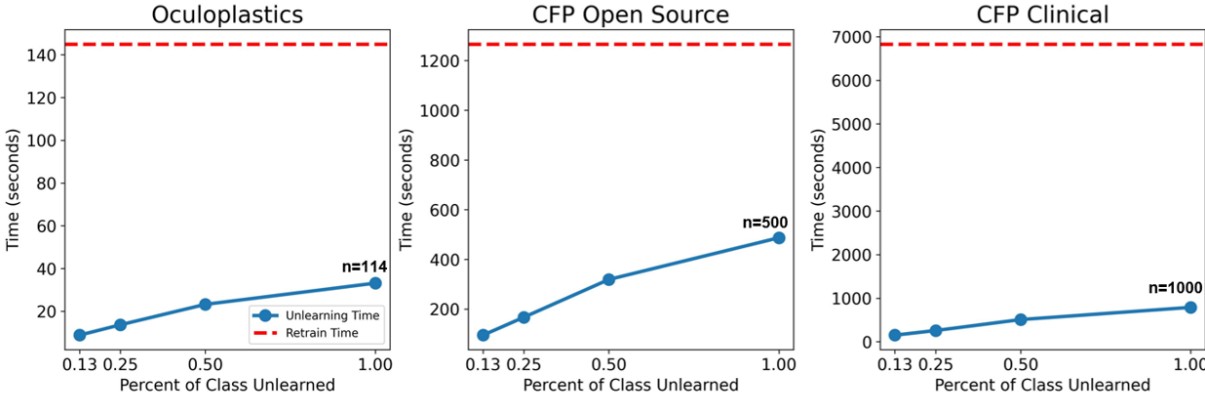

Figure 8: Timing results of unlearning on all clinical datasets evaluated where various percentages of the class to be unlearned where designated as the forget set. n denotes the total size of the class to be unlearned.

### A.7.4   Hyperparameter Tuning

To evaluate the impact of hyperparameters $\gamma$ (noise scale) and $\lambda$ (closeness regularization), we performed grid sweeps and measured accuracy on F and R sets. Columns 1 and 2 of the heatmaps show performance on each subset; Column 3 shows Euclidean distance from retrain baseline, a proxy for closeness to retrain performance (Appendix Figures 9 and 11).

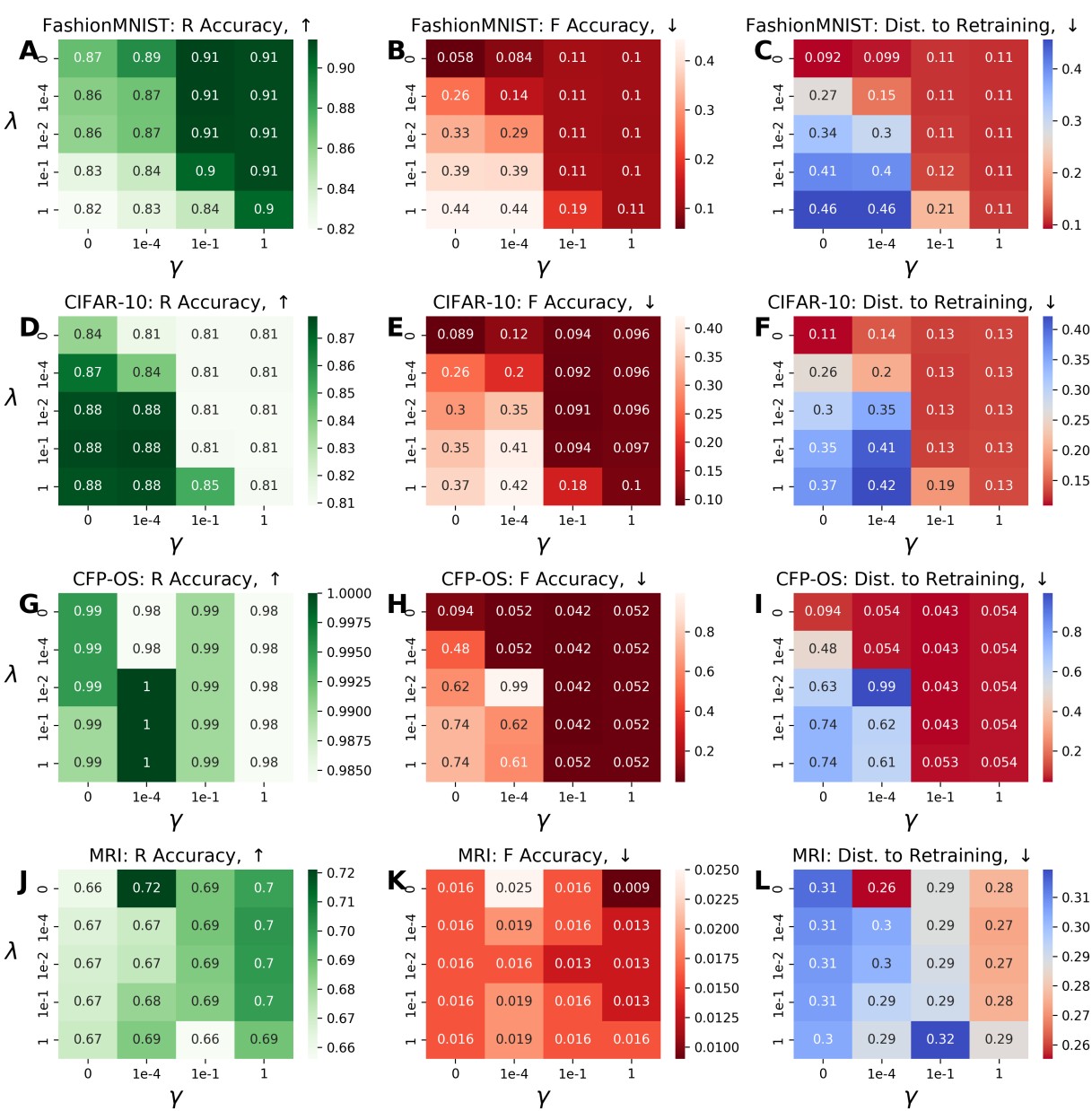

Figure 9: Unlearning results using various combinations of $\gamma$ and $\lambda$ on non-medical imaging datasets. Column 1 denotes the accuracy of all combinations on $R$, Column 2 Denotes the accuracy of all combinations on $F$, and Column 3 denotes the Euclidean distance of $[F_{acc}, R_{acc}]$ obtained using all combinations of $\gamma$ and $\lambda$ from $[F_{acc}, R_{acc}]$ obtained using the retrained model.

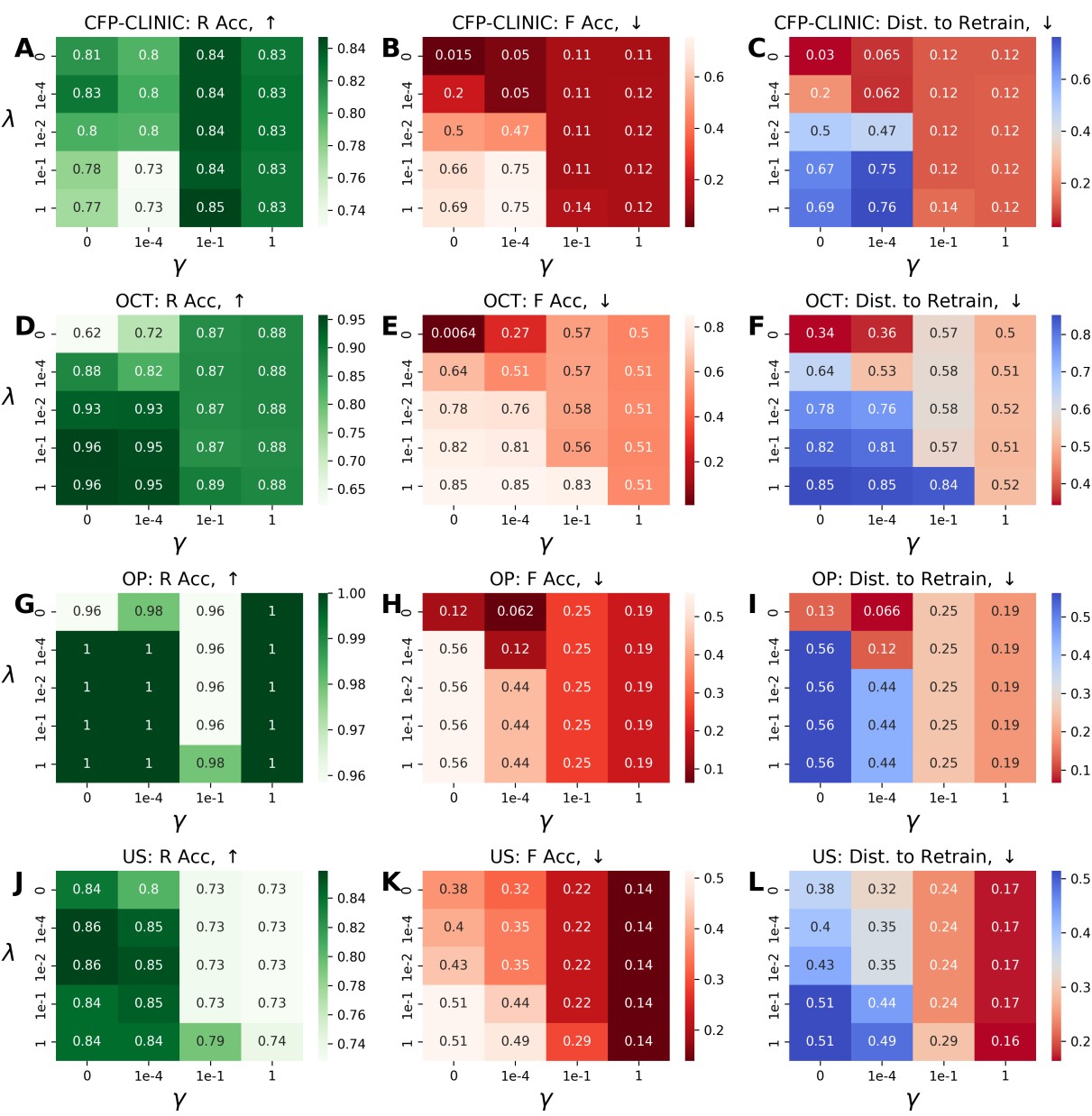

Figure 10: Results of unlearning an entire class using various combinations of $\gamma$ and $\lambda$ on medical imaging datasets. Column 1 denotes the accuracy of all combinations on $R$, Column 2 Denotes the accuracy of all combinations on $F$, and Column 3 denotes the Euclidean distance of $[F_{acc}, R_{acc}]$ obtained using all combinations of $\gamma$ and $\lambda$ from $[F_{acc}, R_{acc}]$ obtained using the retrained model. OP denotes oculoplastic dataset, and US denotes ultrasound dataset. Directionality of arrows indicates whether high or low values are the desired output.

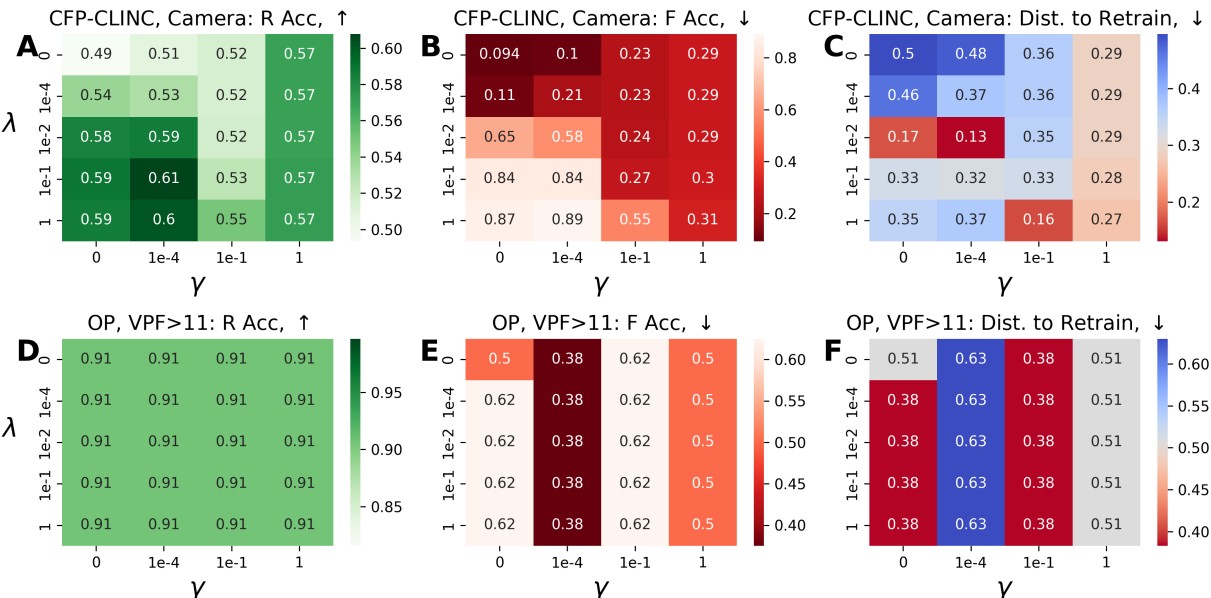

Figure 11: Results of unlearning subsets of images based on clinical parameters using various combinations of $\gamma$ and $\lambda$ on medical imaging datasets. Column 1 denotes the accuracy of all combinations on $R$, Column 2 Denotes the accuracy of all combinations on $F$, and Column 3 denotes the Euclidean distance of $[F_{acc}, R_{acc}]$ obtained using all combinations of $\gamma$ and $\lambda$ from $[F_{acc}, R_{acc}]$ obtained using the retrained model. A-C represents the results of unlearning images acquired with a specific imaging device, and D-F are the results of unlearning images of eyes that had a vertical palpebral fissure $> 11$mm. Directionality of arrows indicates whether high or low values are the desired output.

### A.7.5  Selective Unlearning Raw Data

The following tables provide detailed performance metrics for each forget percentage from 1% to 100%, as used in the selective unlearning plots (Figure 2 and 5). This allows finer inspection of forgetting trends across subset sizes.

## B  Disclosures

S.H.: Stock or stock options e Horizon Surgical Systems. P.S.: Consultant Oyster Point Pharma; Leadership or fiduciary role in other board, society, committee or advocacy group, Board member of American Society of Ophthalmic Plastic and Reconstructive Surgery; Stock or stock options Lodestone Pharmaceuticals. A.Q.T.: Consultant Genetech/Roche. J.C.P, author on provisional patents US20230194734A1, WO2024145070A1 (University of Miami holds all rights)

| Dataset | fashioMNIST | | | CIFAR-10 | | |
|---|---|---|---|---|---|---|
| Metric | R Acc | F Acc | MIA | R Acc | F Acc | MIA |
| Retrain | 0.973 | 0.8 | 0.5 +/- 0.0 | 0.955 | 0.9 | 0.4 +/- 0.2 |
| Finetune | 0.984 | 0.7 | 0.8 +/- 0.245 | 0.957 | 0.8 | 0.6 +/- 0.374 |
| NegGrad | 0.973 | 0.7 | 0.7 +/- 0.245 | 0.933 | 0.5 | 0.6 +/- 0.2 |
| CFK | 0.981 | 1 | **0.5 +/- 0.0** | 0.97 | 1 | 0.7 +/- 0.245 |
| EUK | 0.982 | 1 | 0.7 +/- 0.245 | 0.97 | 1 | 0.5 +/- 0.316 |
| SCRUB | 0.991 | 1 | 0.4 +/- 0.2 | 0.982 | 1 | **0.5 +/- 0.0** |
| Boundary | 0.93 | 0.7 | 0.7 +/- 0.245 | 0.919 | 0.5 | 0.8 +/- 0.245 |
| Ours | 0.93 | 0.7 | 0.9 +/- 0.2 | 0.924 | 0.6 | 0.7 +/- 0.245 |

Table 5: Selective unlearning of all datasets retaining 1% of the forget class. Blue and red highlight denote best and worst performance on $F$ and $R$ accuracy. MIA closest to .5 is bolded.

| Dataset | fashionMNIST | | | CIFAR-10 | | | MRI | | | CFP-Open Source | | |
|---|---|---|---|---|---|---|---|---|---|---|---|---|
| Metric | R Acc | F Acc | MIA | R Acc | F Acc | MIA | R Acc | F Acc | MIA | R Acc | F Acc | MIA |
| Retrain | 0.969 | 0.87 | 0.44 +/- 0.073 | 0.946 | 0.92 | 0.47 +/- 0.081 | 0.958 | 0.92 | 0.552 +/- 0.059 | 0.919 | 0.86 | 0.52 +/- 0.075 |
| Finetune | 0.985 | 0.88 | 0.46 +/- 0.058 | 0.952 | 0.94 | **0.49 +/- 0.058** | 0.965 | 0.975 | **0.491 +/- 0.052** | 0.881 | 0.78 | 0.56 +/- 0.08 |
| NegGrad | 0.965 | 0.81 | 0.61 +/- 0.037 | 0.944 | 0.74 | 0.65 +/- 0.055 | 0.944 | 0.833 | 0.594 +/- 0.068 | 0.902 | 0.82 | 0.52 +/- 0.04 |
| CFK | 0.982 | 0.99 | 0.59 +/- 0.08 | 0.969 | 1 | 0.56 +/- 0.073 | 0.966 | 0.914 | 0.576 +/- 0.033 | 0.916 | 0.84 | 0.72 +/- 0.075 |
| EUK | 0.982 | 0.99 | **0.49 +/- 0.049** | 0.968 | 0.99 | 0.57 +/- 0.081 | 0.967 | 0.975 | 0.509 +/- 0.03 | 0.943 | 0.86 | **0.5 +/- 0.089** |
| SCRUB | 0.99 | 0.94 | 0.46 +/- 0.058 | 0.979 | 0.99 | 0.54 +/- 0.02 | 0.961 | 0.938 | 0.533 +/- 0.056 | 0.895 | 0.86 | 0.48 +/- 0.098 |
| Boundary | 0.908 | 0.42 | 0.76 +/- 0.097 | 0.878 | 0.3 | 0.72 +/- 0.108 | 0.719 | 0.111 | 0.752 +/- 0.07 | 0.758 | 0.36 | 0.7 +/- 0.11 |
| Ours | 0.907 | 0.4 | 0.65 +/- 0.032 | 0.874 | 0.28 | 0.72 +/- 0.06 | 0.729 | 0.191 | 0.691 +/- 0.059 | 0.763 | 0.36 | 0.64 +/- 0.15 |

Table 6: Selective unlearning of all datasets retaining 10% of the forget class. Blue and red highlight denote best and worst performance on $F$ and $R$ accuracy. MIA closest to .5 is bolded.

| Dataset | fashionMNIST | | | CIFAR-10 | | | MRI | | | CFP-Open Source | | |
|---|---|---|---|---|---|---|---|---|---|---|---|---|
| Metric | R Acc | F Acc | MIA | R Acc | F Acc | MIA | R Acc | F Acc | MIA | R Acc | F Acc | MIA |
| Retrain | 0.971 | 0.824 | 0.584 +/- 0.054 | 0.956 | 0.92 | 0.5 +/- 0.061 | 0.943 | 0.931 | **0.494 +/- 0.034** | 0.905 | 0.792 | 0.616 +/- 0.048 |
| Finetune | 0.986 | 0.896 | **0.512 +/- 0.035** | 0.954 | 0.928 | 0.456 +/- 0.069 | 0.952 | 0.928 | 0.553 +/- 0.02 | 0.917 | 0.848 | 0.592 +/- 0.047 |
| NegGrad | 0.96 | 0.684 | 0.66 +/- 0.018 | 0.94 | 0.612 | 0.668 +/- 0.043 | 0.932 | 0.79 | 0.635 +/- 0.054 | 0.924 | 0.792 | 0.688 +/- 0.109 |
| CFK | 0.981 | 0.972 | 0.516 +/- 0.034 | 0.969 | 0.988 | **0.524 +/- 0.023** | 0.97 | 0.938 | 0.565 +/- 0.036 | 0.941 | 0.816 | 0.568 +/- 0.082 |
| EUK | 0.982 | 0.964 | 0.48 +/- 0.052 | 0.969 | 0.992 | 0.576 +/- 0.066 | 0.965 | 0.97 | 0.509 +/- 0.009 | 0.932 | 0.864 | 0.576 +/- 0.125 |
| SCRUB | 0.989 | 0.976 | 0.532 +/- 0.037 | 0.98 | 0.984 | **0.524 +/- 0.045** | 0.954 | 0.958 | 0.459 +/- 0.009 | 0.912 | 0.848 | **0.536 +/- 0.074** |
| Boundary | 0.898 | 0.256 | 0.776 +/- 0.015 | 0.859 | 0.22 | 0.748 +/- 0.085 | 0.675 | 0.049 | 0.78 +/- 0.033 | 0.794 | 0.32 | 0.696 +/- 0.041 |
| Ours | 0.897 | 0.268 | 0.804 +/- 0.029 | 0.86 | 0.208 | 0.784 +/- 0.05 | 0.636 | 0.032 | 0.738 +/- 0.037 | 0.777 | 0.424 | 0.736 +/- 0.041 |

Table 7: Selective unlearning of all datasets retaining 25% of the forget class. Blue and red highlight denote best and worst performance on $F$ and $R$ accuracy. MIA closest to .5 is bolded.

| Dataset | fashionMNIST | | | CIFAR-10 | | | MRI | | | CFP-Open Source | | |
|---|---|---|---|---|---|---|---|---|---|---|---|---|
| Metric | R Acc | F Acc | MIA | R Acc | F Acc | MIA | R Acc | F Acc | MIA | R Acc | F Acc | MIA |
| Retrain | 0.97 | 0.908 | 0.514 +/- 0.058 | 0.949 | 0.864 | 0.526 +/- 0.043 | 0.954 | 0.877 | 0.556 +/- 0.032 | 0.915 | 0.748 | **0.592 +/- 0.097** |
| Finetune | 0.965 | 0.912 | 0.474 +/- 0.05 | 0.945 | 0.924 | 0.494 +/- 0.023 | 0.962 | 0.912 | 0.581 +/- 0.025 | 0.911 | 0.724 | 0.628 +/- 0.035 |
| NegGrad | 0.97 | 0.722 | 0.638 +/- 0.027 | 0.944 | 0.644 | 0.616 +/- 0.051 | 0.898 | 0.749 | 0.568 +/- 0.015 | 0.833 | 0.228 | 0.648 +/- 0.032 |
| CFK | 0.982 | 0.98 | **0.492 +/- 0.032** | 0.968 | 0.978 | 0.504 +/- 0.053 | 0.968 | 0.906 | 0.579 +/- 0.009 | 0.933 | 0.748 | 0.624 +/- 0.032 |
| EUK | 0.982 | 0.968 | 0.472 +/- 0.028 | 0.969 | 0.98 | **0.498 +/- 0.031** | 0.969 | 0.904 | 0.564 +/- 0.024 | 0.927 | 0.792 | 0.612 +/- 0.027 |
| SCRUB | 0.99 | 0.968 | 0.49 +/- 0.03 | 0.981 | 0.982 | 0.548 +/- 0.017 | 0.949 | 0.936 | **0.536 +/- 0.036** | 0.928 | 0.808 | 0.6 +/- 0.038 |
| Boundary | 0.896 | 0.27 | 0.788 +/- 0.049 | 0.852 | 0.158 | 0.786 +/- 0.036 | 0.612 | 0.01 | 0.777 +/- 0.029 | 0.847 | 0.252 | 0.716 +/- 0.015 |
| Ours | 0.898 | 0.292 | 0.766 +/- 0.026 | 0.865 | 0.21 | 0.784 +/- 0.036 | 0.599 | 0.019 | 0.749 +/- 0.057 | 0.837 | 0.236 | 0.696 +/- 0.05 |

Table 8: Selective unlearning of all datasets retaining 50% of the forget class. Blue and red highlight denote best and worst performance on $F$ and $R$ accuracy. MIA closest to .5 is bolded.

| Dataset | fashionMNIST | | | CIFAR-10 | | | MRI | | | CFP-Open Source | | |
|---|---|---|---|---|---|---|---|---|---|---|---|---|
| Metric | R Acc | F Acc | MIA | R Acc | F Acc | MIA | R Acc | F Acc | MIA | R Acc | F Acc | MIA |
| Retrain | 0.967 | 0.796 | 0.577 +/- 0.031 | 0.948 | 0.84 | 0.548 +/- 0.035 | 0.95 | 0.542 | 0.733 +/- 0.011 | 0.939 | 0 | 0.749 +/- 0.036 |
| Finetune | 0.986 | 0.879 | 0.559 +/- 0.014 | 0.956 | 0.907 | 0.473 +/- 0.033 | 0.972 | 0.767 | 0.621 +/- 0.012 | 0.921 | 0.395 | 0.661 +/- 0.051 |
| NegGrad | 0.969 | 0.764 | 0.581 +/- 0.025 | 0.94 | 0.625 | 0.617 +/- 0.03 | 0.947 | 0.36 | 0.749 +/- 0.033 | 0.931 | 0.341 | 0.708 +/- 0.046 |
| CFK | 0.982 | 0.976 | 0.481 +/- 0.02 | 0.968 | 0.987 | 0.553 +/- 0.024 | 0.975 | 0.855 | **0.594 +/- 0.022** | 0.959 | 0.541 | 0.651 +/- 0.057 |
| EUK | 0.981 | 0.953 | 0.468 +/- 0.017 | 0.968 | 0.964 | 0.54 +/- 0.025 | 0.975 | 0.853 | 0.602 +/- 0.013 | 0.965 | 0.56 | 0.641 +/- 0.054 |
| SCRUB | 0.99 | 0.98 | **0.516 +/- 0.014** | 0.982 | 0.979 | **0.529 +/- 0.028** | 0.965 | 0.803 | 0.616 +/- 0.019 | 0.951 | 0.595 | **0.614 +/- 0.068** |
| Boundary | 0.894 | 0.26 | 0.801 +/- 0.014 | 0.867 | 0.165 | 0.775 +/- 0.024 | 0.6 | 0.013 | 0.77 +/- 0.018 | 0.94 | 0.149 | 0.705 +/- 0.049 |
| Ours | 0.896 | 0.277 | 0.784 +/- 0.03 | 0.866 | 0.188 | 0.761 +/- 0.038 | 0.648 | 0.011 | 0.78 +/- 0.022 | 0.954 | 0.165 | 0.671 +/- 0.075 |

Table 9: Selective unlearning of all datasets retaining 75% of the forget class. Blue and red highlight denote best and worst performance on $F$ and $R$ accuracy. MIA closest to .5 is bolded.

| Dataset | fashionMNIST | | | CIFAR-10 | | | MRI | | | CFP-Open Source | | |
|---|---|---|---|---|---|---|---|---|---|---|---|---|
| Metric | R Acc | F Acc | MIA | R Acc | F Acc | MIA | R Acc | F Acc | MIA | R Acc | F Acc | MIA |
| Retrain | 1 | 0 | 0.81 +/- 0.08 | 0.975 | 0 | 0.876 +/- 0.022 | 0.899 | 0 | 0.943 +/- 0.011 | 0.942 | 0 | 0.945 +/- 0.019 |
| Finetune | 1 | 0 | 0.84 +/- 0.066 | 0.983 | 0 | 0.816 +/- 0.047 | 0.889 | 0.921 | 0.514 +/- 0.013 | 0.938 | 0.862 | 0.552 +/- 0.025 |
| NegGrad | 0.969 | 0 | 0.83 +/- 0.121 | 0.983 | 0.094 | 0.756 +/- 0.04 | 0.903 | 0.557 | 0.635 +/- 0.021 | 0.942 | 0.649 | 0.67 +/- 0.035 |
| CFK | 0.99 | 0 | 0.84 +/- 0.073 | 0.987 | 0.597 | 0.684 +/- 0.051 | 0.906 | 0.914 | **0.5 +/- 0.01** | 0.939 | 0.911 | **0.519 +/- 0.021** |
| EUK | 0.984 | 0 | 0.81 +/- 0.066 | 0.986 | 0.666 | **0.625 +/- 0.02** | 0.905 | 0.926 | 0.502 +/- 0.024 | 0.94 | 0.894 | 0.531 +/- 0.018 |
| SCRUB | 0.995 | 0 | 0.75 +/- 0.055 | 0.973 | 0.322 | 0.744 +/- 0.029 | 0.913 | 0.913 | 0.484 +/- 0.017 | 0.944 | 0.897 | 0.57 +/- 0.028 |
| Boundary | 0.87 | 0.058 | 0.8 +/- 0.095 | 0.84 | 0.089 | 0.756 +/- 0.038 | 0.66 | 0.016 | 0.808 +/- 0.031 | 0.99 | 0.094 | 0.826 +/- 0.015 |
| Ours | 0.89 | 0.084 | **0.68 +/- 0.068** | 0.81 | 0.094 | 0.703 +/- 0.06 | 0.7 | 0.009 | 0.797 +/- 0.021 | 0.99 | 0.042 | 0.854 +/- 0.005 |

Table 10: Selective unlearning of all datasets retaining 100% of the forget class. Blue and red highlight denote best and worst performance on $F$ and $R$ accuracy. MIA closest to .5 is bolded.

