# OpenReview forum: "Targeted Unlearning Using Perturbed Sign Gradient Methods With Applications On Medical Images"
_TMLR — Accepted by TMLR_

### Review · Reviewer_px6p · 2025-09-10

**Summary Of Contributions:**

This work targets machine unlearning by a boundary-based unlearning mechanism. The authors formulate a compelling motivation from the perspective of practical application within the healthcare sector (medical imaging). They position machine unlearning to go beyond compliance to legislation (General Data Protect Regulation (GDPR), and similar) casting it more similar to continual learning. In particular , their proposed mechanism, but also machine unlearning in general, can be understood as a tool for revising models to improve performance, yet form the perspective of removing certain information, in contrast to adding it.  This perspective allows for selectively removing information, such as data from outdated scanners or low-quality images, to improve model relevance and performance.

The core contributions include the proposed algorithm itself, a theoretical analysis providing convergence guarantees, and an empirical evaluation on both standard and newly introduced benchmarks.

**Strength:**

1. The authors address machine unlearning beyond a compliance perspective, motivated from real-world applications in healthcare.
2. The authors purposefully selected a optimization formulation that allows them mathematical introspection of convergence properties of their method.
3. The introduction of new benchmarks tailored to their proposed use cases is a valuable contribution that could foster further research in this specific application area. (If the data is made available to the community)

**Weakness:**

My main concerns revolve around a core definition of the unlearning objective, namely: “C1: the updated model should misclassify $(x,y) \in F$, i.e. $f_{w_u}(x) \neq y$”. It reads to misclassifying as many samples in the forget set as possible, which is not the default criteria, and brings various implications to the evaluation of the proposed method, which to me, are not discussed sufficiently.

**1. Conceptual Formulation of the Unlearning Goal:**
I found the formulation of the unlearning goal, particularly desideratum **C1**, to be at odds with the standard definition in the literature. The authors state that the updated model should misclassify samples from the forget set F (i.e. $f_{w_u}(x) \neq y$,), which is measured by Forget Set Accuracy (F-Acc), where lower is better.
However, the common objective of unlearning is to produce a model that is indistinguishable from one that was retrained from scratch on the retain set [1,2,3]. This means the model's accuracy on the forget set should be similar to that of a retrained model, which is typically close to random chance for a balanced dataset, but not necessarily zero. This (standard) view is articulated in key papers the authors cite, such as Triantafillou et al. [1] and Golatkar et al. [2].

To me, this mismatch in definition impacts the interpretation of the results throughout the paper. For instance, the claim in Section 4.1 that "Our method consistently achieves lower F-Acc than all baselines" is presented as a key advantage. However, under the standard unlearning paradigm, this might indicate that the method is indeed memorizing the forget set rather forgetting it.
Further, it comes with important albeit more conceptual implications I will discuss later below (see **Broader impact concerns**.).

**Suggestion:** I would greatly appreciate it if the authors to further clarify and motivate the desideratum C1.  If their goal is indeed to force misclassification, this represents a significant departure from the standard unlearning paradigm and would require a thorough discussion to justify this choice and differentiate it from prior work. Alternatively, if I have misunderstood, a clearer explanation in the text should resolve this ambiguity.

**2. Statement on Retraining:** Related to the point above, I am confused by the statement in Section 4.2.2: “Retraining may not unlearn. Importantly, we note that retraining alone did not result in significant misclassification of the forget set […]”. By definition, retraining from scratch is the gold standard for unlearning, right? The fact that it doesn't lead to "significant misclassification" (i.e., near-zero accuracy) highlights the potential discrepancy regarding the definition of unlearning in this work.

**3. Clarity of claim “Reliable Behavioral Control”: T**he paper claims the proposed algorithm offers "tunable loss design" and "reliable behavioral control.". However, the ablation study in Section 4.3 concludes that "Seemingly small adjustments can produce large performance shifts.". While the authors commendably report this sensitivity, yet it seems to contradict the idea of "reliable control.”

**Suggestion:** Could the authors provide more nuance to this claim? To substantiate the claim of "tunability," it would be helpful to include a discussion on how a user might intuitively or heuristically select hyperparameters for a desired outcome without an exhaustive grid search. Potentially corroborated by respective experiments.

**4. Clarity of claim “Compositionality”:**  The motivation and evaluation of the model composition approach in Section 4.4 feels underdeveloped. The particular motivation from observing hyperparameter sensitivity to proposing model composition as a solution is not immediately clear.

**Suggestion:** This section would be strengthened by a more detailed conceptual discussion and more thorough empirical evaluation. For instance, what is the intuition behind what composition achieves at the model level? What about other recently well established compositions such as weight-space merging (a.k.a. model merging, model fusion)?
Also, the evaluation in Section 4.4.2 would benefit from the baseline of comparing the composed model to a single, non-composed model that is simply fine-tuned on the retain set.

**5. MIA:** Given the significant role of Member-Inference Attack (MIA)  in the evaluation, MIA could be formally introduced as a desideratum in Section 2.1. Additionally, the MIA scores for the proposed method appear relatively high in some experiments, and a discussion interpreting these results would be welcome.

**6. Is data removal beneficial?** For the use case of forgetting acquisition devices, the paper would be more compelling if it first demonstrated that retaining data from an older device actively harms the model's performance on data from newer devices. Otherwise, more intuition or discussion could clarify why the particular data should be removed from the training set in the first place. This would establish a stronger baseline and motivation for the unlearning task.

**Minor Comments:**

**Related Work:** A dedicated "Related Work" section would greatly help readers who are less familiar with the field to better situate this paper within the broader landscape of unlearning and model editing approaches.

**Comparison to Chen et al.:** The introduction states that Chen et al. "lacks convergence guarantees," but the paper later notes that Chen et al. is a special case of the proposed method. This implies that the convergence guarantees presented here might extend to their work. Clarifying this point would be helpful.

**Clarity in Section 2.2:** The opening sentence, “*Because post-hoc unlearning must modify a frozen model [...] by modifying parameters $w$*" is slightly confusing. If parameters  are being modified, the model is not "frozen." A rephrase could improve clarity.

**Figure 2 Labels:** The x-axis labels in Figure 2 seem to be in the wrong unit, it reads 0.5% instead of 50% as mentioned in the text.

**Intuition for Lemma 2.1:** The explanation could be more explicit. The sentence "we apply via standard application of the Karush-Kuhn-Tucker (KKT) conditions" seems to miss a word. Please clarify what is being applied or shown.

**Conceptual Difference from Prior Work:** The paper would benefit from a brief discussion on the conceptual difference between this method and the previous boundary-based method by Chen et al., particularly regarding the decision to not move the point on the decision boundary. Intuitively I can imagine there are trade-offs to be made? Those can be discussed.

[1] (Triantafillou et al.) - "Are we making progress in unlearning? Findings from the first NeurIPS unlearning competition"
[2] (Golatkar et al.) - "Eternal sunshine of the spotless net: Selective forgetting in deep networks."
[3] (Xu et al.) - "Machine Unlearning: A Survey"

**Audience:**

Yes

**Audience Explanation:**

Machine unlearning recently received a lot of attention, and the general positioning of this work for medical imaging in itself is interesting.
Given that my doubts for regarding the support for the claims of this paper would be addressed, I am sure there will be an audience that is interested in the findings of this work.

**Broader Impact Concerns:**

How I understand the text, forgetting is equated with “successful” misclassification of the forget set samples.
The authors do not discuss the potential negative consequences for the patients, that is: If a patient invokes their right to be forgotten under this proposed method, does the resulting model now carry a detrimental bias targeted at that person? Does the model actively misdiagnose this person in the future? This possibility of inducing targeted failures in a high-stakes domain like medical imaging constitutes a serious ethical risk.

**Claims And Evidence:**

No

**Claims Explanation:**

While the paper provides empirical results and theoretical derivations, I find that several of the main claims are not convincingly supported by the evidence presented, due to the following key reasons:

1. The primary claim of superior unlearning performance rests on a definition (forcing misclassification on the forget set) that diverges significantly from the established consensus in the literature. Because this foundational evaluation criterion is not sufficiently justified, the subsequent results (e.g., achieving lower 'F-Acc') are not convincing as evidence of successful unlearning as it is commonly understood.
2. Regarding the claim of "reliable behavioral control", the paper's own ablation study highlights the method's high sensitivity to hyperparameter changes ("small adjustments can produce large performance shifts"). This evidence supports the idea of tunability, but not the reliability, or predictability, which are key to the claim.
3. Regarding the claim of "compositionality", to me, the motivation for this approach is not well-established in the work. I would require at least more intuition to what the composition (stitching) of models means on a conceptual level. Also, the empirical evaluation misses baselines, e.g. the non-composed finetuned on the remain set, to make a convincing case for its utility.

Importantly, it is to note that the theoretical claim regarding convergence guarantees does appear to be well-supported. My concerns lie primarily with the interpretation and strength of the empirical evidence backing the practical claims of the method.

**Requested Changes:**

1. **Address concerns regarding the unlearning objective and evaluation:** As this is my most significant concern, the authors would need to address the conceptual mismatch between their definition of unlearning (forcing misclassification) and the standard definition in the literature (approximating a retrained model's accuracy). I see two possible paths forward:

    Either, re-evaluate the method using standard unlearning metrics. This would involve comparing the unlearned model's performance on the forget set to that of a model retrained from scratch, rather than aiming for zero accuracy. This would likely require re-running experiments and re-framing the results.

    **Or,** add a sufficient justification. If the authors maintain their definition, they must add a substantial discussion that explicitly acknowledges the departure from the standard paradigm. This section must rigorously justify why forcing misclassification is a more desirable or appropriate goal for their target applications and discuss the potential drawbacks of this approach.

2. **Revise Claims on "Reliable Control" and Tunability:** The claims of "reliable behavioral control" are currently not supported by the evidence of high hyperparameter sensitivity.

    I would advise to provide further nuance to these claims. The authors should discuss the challenges of tuning the method and provide heuristics or guidance for practitioners on how to select hyperparameters for a desired outcome. Best supported by respective experiments.

3. **Strengthen or Remove the "Compositionality" Section (4.4):** As it stands, to me, the claims of this sections are not convincingly demonstrated.

    To keep this as a contribution, the section requires expansion. This includes a clearer motivation, a more in-depth conceptual explanation to the composition mechanism, and a thorough empirical evaluation that includes baselines, such as comparing the composed model to a single, non-composed, model fine-tuned on the retain set. Otherwise, I would recommend removing this section to strengthen the paper's focus.

4. **Additional motivation for device forgetting:** The experiment on forgetting acquisition devices needs a baseline which demonstrates that not forgetting the data from the old device harms the model's performance. This would clearly establish the motivation for this task. Alternatively, further motivation to why the “outdate” data should be forgotten non-the-less should be provided.

5. Slightly enhance clarity as pointed out in the minor weaknesses above (Summary Of Contributions).

---

> ### Author Response · Authors · 2025-10-09
> **Clarification on Unlearning Definition**
>
> Q: The formulation of the unlearning goal (C1) as forcing misclassification is at odds with the standard definition, which measures success by similarity to retraining which impacts the interpretation of results.
>
> We thank the reviewer for this important conceptual question! We agree that our formulation departs from the standard definition of unlearning, which evaluates success by statistical indistinguishability from retraining on $R$. Our approach instead operationalizes unlearning for model maintenance, where the goal is to actively suppress the influence of designated forget samples.
>
> While distinct, our formulation is not contradictory to the retraining-based definition. To see this, if one desires random-like predictions on F, our procedure can be easily adapted as follows: 1. unlearn random fractions of F (ie, 1/k subsets), 2. use the ensemble of unlearned models to obtain random classification on F. In this way, the reviewer will see that while our definition is slightly different, it is not contradictory.
>
> Revision: To this end, we have clarified this in Section 2.1 and framed our definition as complementary to the retraining-based paradigm. We have also added a new Remark 2.1 comparing and contrasting our definition of unlearning as defined by C1 with the fields, as well as added a short discussion on the difference in the final paragraph of the submission.
>
> Second, our goal in this work is motivated by unlearning as a tool for selective removal of samples with applications for clinical model maintenance. As such, we define successful forgetting (C1) as high misclassification on the designated Forget set samples.  We do not consider this as memorization; rather than fitting F more tightly, our boundary-shifting procedure deliberately pushes F away from its original labels, ensuring those signals do not remain usable.
>
> Revision: We have clarified this point in Section 2.1 of the revised manuscript.
>
> Finally, the reviewer may verify that our use of Forget Set Accuracy (F-Acc) is in line with prior work such as Golatkar et al. 2020, Triantafillou et al. 2023, Chen et. al. 2023 with F as a key metric. In our loss, we explicitly interpret this metric as a practical operationalization of forgetting for clinical maintenance scenarios. While retraining is the gold standard for privacy-driven unlearning, it does not provide explicit control over outcomes on F. When F contains samples representative of their class, retraining can still yield high accuracy on those samples (see Tables 9–10 in our work and Table 1 in Golatkar et al. 2020). By contrast, our method provides a mechanism to explicitly suppress such contributions which is relevant in our applications as new medical conditions are identified by the community.
>
> Q: Confusion regarding the statement in Section 4.4.2 that ‘retraining may not unlearn’, and how this again may be at odds with our stated formulation of unlearning, especially as retraining from scratch is widely considered the gold standard.
>
> We agree that retraining from scratch is widely considered the gold standard for unlearning. Our statement that “retraining may not unlearn” was not intended to challenge this consensus, but to highlight a nuance relevant under our definition of forgetting. Retraining ensures that the model has no direct exposure to F, but it does not provide explicit control over how F is treated after retraining. In cases where F contains samples that are highly representative of their class, a retrained model can still classify them correctly which will result in a high accuracy value. We observe this in our empirical analysis, as in Tables 9–10 where retraining with a fraction of an entire class did not induce low accuracy on the forget set. By contrast, our boundary-shifting approach allows practitioners to explicitly enforce misclassification on F.
>
> Revision: We have added a paragraph in the results section to clarify this nuance and to avoid overstatement.
>
> `Retraining and unlearning. Retraining from scratch is widely regarded as the gold standard for unlearning, as it guarantees that the model has no direct exposure to the forget set. However, in our experiments (Figure 3, Table 1), we observed that retraining did not always lead to low accuracy on $F$. This is expected when $F$ contains samples that are highly representative of their class, since the model can still rely on similar examples present in the remain. Our method differs in that it provides an explicit mechanism to suppress predictive influence from $F$, offering direct control over the unlearned model.`

---

> ### Author Response · Authors · 2025-10-09
> **Response to comments about model compositions and ablation studies**
>
> Q: Concerns about the robustness of the compositionality section of the manuscript (Section 4.4). The authors may consider removal of this section.
>
> We originally included this section in order to leverage multiple unlearned models following the ablation study. However, we see the reviewer’s concern that as it stands it may distract from the main messaging of our manuscript.
>
> Revision: We have removed Section 4.4.
>
>
> Q: Clarification needed on reliability claims and practical guidance for algorithm setting selection, given sensitivity observed in ablation studies
>
> Thank you for pointing this out.
>
> Revision: In the revised manuscript, we now emphasize tunability rather than reliability (see Abstract and Section 1). In the new Section 4.3 (Ablation Studies), we clarified that the framework is sensitive to the modifications in the optimization process, but follows systematic and interpretable patterns: using logits from the forget set enforces strong forgetting at the expense of retention and incorporating remain loss as logits improves retention while reducing forgetting.
>
> To incorporate the reviewer’s suggestion, we added a dedicated paragraph in the Discussion providing practical heuristics for parameter selection. Specifically, we advise (i) using argmax labels with no remain loss for maximum forgetting, (ii) incorporating remain loss with moderate weight for balanced retention. We also explicitly acknowledged the challenge that small changes in the algorithm settings can produce large shifts in high-stakes or resource-constrained settings, and highlight the development of more robust selection strategies as an important direction for future work. With respect to the influence of hyperparameters (such as lambda), we note that modifying lambda in the ablation experiments produced a minimal effect on F and R accuracy. To illustrate this, we have made updated the ablation table in the main text to include a wider range of lambda values.
>
>  We hope that these changes provide nuance to our original claim and give practitioners actionable guidance while avoiding overstatement.

---

> ### Author Response · Authors · 2025-10-09
> **Response to whether or not MIA should be a formal desideratum**
>
> Q: Should MIA be a formal desideratum? Additionally, MIA values are high in some experiments
>
> Indeed, MIA is a valuable unlearning metric for evaluation purposes. However, our work is primarily motivated by clinical model maintenance rather than privacy compliance. So we have not included MIA as a core desideratum in our framework. In Section 2.1, we have clarified that MIA is treated as an evaluation metric reported alongside C1 (forgetting), C2 (retention) for completeness and comparison to prior work. We will include a formal desideratum for improved clarity.
>
> In addition, we have expanded the Discussion to address our MIA results. We note that our method sometimes yields higher MIA scores than retraining or other baselines. We believe that this may be due to the fact that explicit boundary-shifting may leave sharper or non-smooth decision regions around forget samples, which can be exploited by membership inference attacks. Reviewer will note that prior boundary-based methods (e.g., Chen et al. 2023) did not report MIA experiments, so our observations highlight an interesting tradeoff that warrants further investigation. Integrating MIA robustness into our formulation to be an important direction for future work in privacy-focused applications.

---

> ### Author Response · Authors · 2025-10-09
> **Response to concern regarding removal of imaging device baselines and broader ethical concerns**
>
> Q: There is no evidence that inclusion of certain imaging devices harms performance, therefore it seems like a baseline may be missing.
>
> Our current experiments do not show that data from the Cirrus 800 FA device is intrinsically harmful to model performance. However, we do not want to make this claim. In this work, our main goal is to provide proof-of-principle demonstration of how unlearning can be applied in clinically realistic maintenance scenarios. To our knowledge, this aspect of unlearning has not been explored in the applications literature. In practice, medical imaging systems frequently face distribution shifts when equipment is upgraded or replaced, and such shifts are well-documented causes of performance degradation if not addressed (see citations added in the Discussion: Guan et al. 2021, Kumari et al. 2024).
>
> Revision: We have revised the Discussion to clarify that our device-forgetting experiment is designed to illustrate feasibility and mechanism, not to assert that this specific device negatively impacts performance. Future work could more directly quantify performance tradeoffs between device retention and removal.
>
> Q: Broader Ethical Concerns.
>
>
> The reviewer has raised an extremely thoughtful and important point. We agree that defining forgetting as enforced misclassification carries potential risks in high-stakes domains such as healthcare, especially if applied without safeguards. As the reviewer notes, unlearning under our formulation could induce targeted errors for patients who invoke their right to be forgotten or for subgroups removed for maintenance reasons. We are grateful to the reviewer for encouraging us to make these considerations explicit, as they strengthen the framing of our work and clarify its intended scope.
>
> Revision: We have added a new section at the end of the paper, Implications of Unlearning in Healthcare, where we explicitly discuss these concerns. We emphasized that our contribution is methodological rather than prescriptive for immediate clinical deployment, and highlighted the need for safeguards such as registries to track which patients or subgroups have been unlearned, governance mechanisms to prevent inappropriate diagnostic use of unlearned models, and oversight to mitigate potential bias introduction.

---

> ### Author Response · Authors · 2025-10-09
> **Response to Minor Concerns**
>
> Q: The paper would benefit from an explicit Related Work section.
>
>
> In the revised manuscript, we have added a clearly labeled Related Work paragraph at the end of the Introduction. This section summarizes prior approaches to parameter-modification unlearning, including Fisher Information Matrix–based scrubbing \cite{golatkar2020eternalsunshinespotlessnet}, gradient-based approximations \cite{peste2021ssse}, and Hessian approximations \cite{mehta2022deep}, as well as the decision-boundary method of Chen et al. \cite{chen2023boundary}. We then explicitly position our contribution as an extension of boundary-based unlearning into a bilevel optimization framework with convergence guarantees. We hope this modification serves to better situate our work in the context of the field.
>
>
> Q: Clarity in Section 2.2: The opening sentence, “Because post-hoc unlearning must modify a frozen model [...] by modifying parameters " is slightly confusing.
>
> We appreciate this catch as the wording was confusing. We have changed the word ‘frozen’ to ‘learned’ to better denote the true experimental setup.
>
>
> Q: Figure 2 Labels: The x-axis labels in Figure 2 seem to be in the wrong unit, it reads 0.5% instead of 50% as mentioned in the text.
>
> Great catch and much appreciated! In the revised manuscript, we updated the x-axis of Figure 2 (and Figure 5 for consistency) so that the tick marks now read as percentages (0%, 25%, 50%, 75%, 100%) and the axis title is labeled “Forget %.” This resolves the unit ambiguity and aligns the figures with the descriptions in the text.
>
>
> Q: Intuition for Lemma 2.1: The explanation could be more explicit. The sentence "we apply via standard application of the Karush-Kuhn-Tucker (KKT) conditions" seems to miss a word. Please clarify what is being applied or shown.
>
> We have removed the words ‘via standard application’ to more clearly denote that we apply KKT conditions on the inner minimization problem
>
> ‘In essence, we apply Karush-Kuhn-Tucker (KKT) conditions \cite{nocedalwright} on the inner minimization problem.’;
>
>
> Q: Do the presented convergence guarantees extend to Chen. et. al.?
>
> Our convergence guarantees apply specifically to the proposed perturbed sign-gradient inner optimization scheme analyzed in Theorem 2.5 (Section 2.3). The Boundary Shrink algorithm of Chen et al. can be viewed as a special case of our framework obtained by removing the perturbation and regularization terms (or setting the weights to 0). To our knowledge, the convergence properties of the update rule in Chen et al is still unknown. Our key idea is to provide global convergence guarantee for an update rule similar to Chen et al by using a limited amount of noise. Accordingly, while our formulation unifies their algorithm at the implementation level, the convergence proof pertains only to our variant of the boundary shrinking algorithm. We have clarified this in Section 2.3.
>
>
> Q: The paper would benefit from a discussion of the conceptual difference between the new method and chen et.al. particularly wrt to not moving the point on the boundary.
>
>
> Our approach extends the boundary-based intuition of Chen et al. but differs in how boundary points are determined and leveraged. There are two key differences: 1. Chen et al. use a single fixed FGSM-style step to approximate a boundary crossing, whereas we cast this as an explicit inner optimization problem solved by a perturbed sign-gradient update, 2. We incorporate regularization in our formulation which is not present in Chen et al. Intuitively, regularization keeps our update rule slightly conservative compared to large updates in Chen et al’s update rule. We have added Remark 2.6 (Section 2.3)  to briefly discuss the differences between our approach and Chen et. al.

---

> ### Comment · Reviewer_px6p · 2025-10-16
> **The rebuttal clarified the authors' proposed method. However, this clarification has raised new questions about the practical use-case and intended audience for the work**
>
> I thank the authors for their detailed rebuttal. I believe the rebuttal has clarified the proposed method for me. To be sure, my initial understanding of "unlearning" was from a privacy-preserving viewpoint, where the model should act as if it had never been trained on the specific data. My understanding now is that the authors define "forgetting" as a targeted reduction in accuracy for certain samples. With this distinction in mind, I recognize that the manuscript presents a mechanism for such "targeted unlearning" and provides theoretical and empirical evidence to support its working.
> For the benefit of the reader, I would recommend to make this distinction clear early in the paper to avoid potential misunderstandings.
>
> **Target Audience:** If I am indeed understanding the proposed targeted unlearning mechanism better now, the rebuttal has led me to question the practical application and intended audience for the proposed mechanism. Admittedly, I struggle to construct a use-case where targeted unlearning as proposed in this work will be intended and I hope the authors can help me. Particularly, as is, I do not agree with this claim in the abstract: *“This work establishes machine unlearning as a modular, practical alternative to retraining for real-world model maintenance in clinical applications”*.
>
> In this regard, I don’t yet see the evidence in the manuscript. Rather, the choice of the medical domain now seems puzzling. On the one hand, the "maintenance" use-case is not convincing to me — the manuscript does demonstrate the ability to misclassify (forget) selected samples while (almost) preserving the performance on the retain set, but a compelling, practical, scenario where this capability serves a requirement remains unclear to me. The authors do mention that *"medical imaging systems frequently face distribution shifts when equipment is upgraded or replaced"*, however, this seems like the classical problem setting of continual learning. How will the targeted forgetting will aid this issue? To my understanding this has not been demonstrated in the current version of the paper.
> On the other hand, the ethical implications of applying the proposed method in the medical field are significant. Choosing a different domain, such as industrial defect-detection, might have circumvented some of these ethical concerns.
>
> I am afraid, I would need the authors to convince me of a use-case where the proposed method is a considerable option. At best, I would like to see that use-case evaluated.
>
> **Regarding the tunability of the proposed method:** The manuscript offers empirical intuition into the effects of certain configurations of the introduced hyper-parameters, and, it also acknowledges the method's sensitivity to these hyper-parameters. While deferring a deeper exploration to future work is of course valid, it does take from the contribution of the current manuscript. I am torn to acknowledge the tunability itself as contribution, as it seems to be a property of introducing any hyper-parameter.
>
> **Regarding the newly added related work section:** While I do appreciate the authors' effort of adding a related work section, to me, the integration of this work into the related field is not easy for the reader to grasp. As is, I understood that different methods use differently compute expensive approximations of parameter importance to “dynamically compute the appropriate noise to scrub the parameters of a Deep Neural Network of the Forget set”. The difference to the method proposed by Chen et al. is rather implicit as is the connection to this work.
> I would recommend to shift focus on the most prominent conceptual differences of the proposed method in comparison to the established literature, and point to a survey for details of the individual methods. If the authors wish to elaborate some methods in more detail, the whole section could move to the appendix as well.
>
> **Regarding “compositionality":** The authors choose to remove Section 4.4 which was discussing the compositionality aspect of their method. However compositionality is still mentioned in the abstract, in introduction to the results section, and Figure 8 in the appendix (which is also now not referred to anymore in the manuscript).

---

> > ### Comment · Reviewer_px6p · 2025-10-16
> >
> > **Regarding “difference in unlearning goal from related work to this work”:** By adding the nuance to the use of "forgetting" as done currently, I feel the text became slightly self-contradicting and could benefit from another iteration.  The authors maintained the passage: *“Following prior work, we define unlearning success via two desiderata on model […]”*, and follow-up in the next paragraph with: *“We note that our formulation of desideratum (C1) which requires misclassification of the forget set differs from the standard definition of machine unlearning, […]”*. In the rebuttal answer it was further clarified that: *“While distinct, our formulation is not contradictory to the retraining-based definition. To see this, if one desires random-like predictions on F, our procedure can be easily adapted as follows: 1. unlearn random fractions of F (ie, 1/k subsets), 2. use the ensemble of unlearned models to obtain random classification on F. In this way, the reviewer will see that while our definition is slightly different, it is not contradictory.”*
> > I still do see a significant difference in the definitions (as discussed above) and would like to raise two of my issues with the rebut: Firstly, as the authors state themselves in the manuscript, it is not about “random-like predictions”, it is about “predictions as obtained from the retain set”. Secondly, “easily adapting” sounds a bit overstated to me. It could also be seen as whole new mechanism, which then needs further evidence, at least a citation for its working. And why would one want “random-predictions” in a targeted-unlearning setting? Of course, this is taking the privacy-related perspective to unlearning again, and hence besides the point for this work. I chose to keep nagging on this point because I hope the discussion might help the authors to navigate readers like me in a next iteration of this work.
> >
> > **Minor observation:** Figure 2 seems to have deteriorated in resolution in the new version of the manuscript.

---

> > > ### Author Response · Authors · 2025-10-23
> > > **Real world use cases, tunability, related work, residual text**
> > >
> > > Q4: What are potential real-world use cases of unlearning for model maintenance in medical settings?
> > >
> > > While our manuscript focuses on two maintenance cases (device deprecation and phenotype exclusion), there are several broader clinical situations where selective removal of information may be advantageous:
> > >
> > > 1. Policy and consent updates: If certain data becomes non-compliant through policy changes, unlearning offers a means to withdraw their influence without full model retraining.
> > > 2. Demographic recalibration: As demonstrated in our experiments, unlearning can suppress overrepresentation of specific subgroups or phenotypes.
> > > 3. Quality-control and error correction: When systematic labeling and/or acquisition errors are discovered retrospectively, unlearning can remove their residual effect on model parameters without requiring complete retraining.
> > > 4. Adaptation of foundation models: For large-scale pretrained models released for general medical use, unlearning could enable rapid domain steering when local retraining or fine-tuning is impractical and there is access to the original training data.
> > >
> > > These examples align with the fidelity use case of machine unlearning described in prior surveys of machine unlearning (Nguyen et al 2024 ACM Computing Surveys), where unlearning serves to maintain data integrity rather than privacy. Kurmanji et al NeurIPS 2023 also note that unlearning use cases are inherently application-specific; our work extends this argument to clinical maintenance contexts by providing concrete demonstrations on real-world imaging data.
> > >
> > > While these are possible applications, they are still conjectures in many respects. Of course, such applications of machine unlearning would also require well agreed upon ethical regulations and governing body oversight which we discuss in Section 6 of the revised submission.
> > >
> > >
> > > Q5: Is the tunability a contribution?
> > >
> > > Our intent was not to present the tunable elements as generic hyperparameters, but rather as structured or modular extensions of the core algorithm with controllable trade-offs between forgetting and retention. Specifically, the three introduced heuristics are design choices within the optimization procedure, not independent hyperparameters. In the revised manuscript, we clarified this distinction by explicitly referring to them as configurable extensions rather than tunable hyperparameters (end of section 2.4), and by summarizing their effects empirically in Section 4.3.
> > >
> > > Separately, the regularization weight $\lambda$, which is the only hyperparameter (as $\gamma$ was fixed in these experiments), was also explored in ablation experiments but is conceptually distinct from these algorithmic settings. We hope this revision clarifies that our contribution lies in providing and systematically analyzing these structured extensions.
> > >
> > >
> > > Q6: Regarding the newly added related work section
> > >
> > > We thank the reviewer for this suggestion. In the revised manuscript, we have added citations to comprehensive surveys of machine unlearning (Xu et al 2023 and Nguyen et al 2025). We will further expand the related work section for the camera-ready version if needed.
> > >
> > >
> > > Q7: Residual text on compositionality
> > >
> > > We thank the reviewer for catching this oversight. In the revised manuscript, we have removed all remaining references to compositionality, including those in the abstract, the introduction to the results section, and Figure 8 in the appendix (which has been deleted). We confirm that the concept is no longer mentioned anywhere in the text or figures to ensure consistency with the current manuscript structure.

---

> > > ### Author Response · Authors · 2025-10-23
> > > **On the difference in unlearning goals and minor issue**
> > >
> > > Q8: Concerns on the text on difference in unlearning goals from our work to related work.
> > >
> > > We thank the reviewer for bringing this up again as it has sparked meaningful discussions both here and internally related to our work. We agree that the text in the revision was vague. To address this, we have removed the phrase ‘following prior work’ and modified the beginning of the second paragraph to note that it is `slightly different` from the standard definition of unlearning in Section 2.1 and Remark 2.1. Furthermore, we agree that random-like predictions are not required in our use case, and we have made this explicitly clear now in Remark 2.1.
> > >
> > > Regarding the reviewer’s second concern, the discussion of ‘easily adapting our method to the ensemble approach’ appeared only in our previous rebuttal and was intended as an intuition rather than a formal argument or proof. We agree that positioning this argument through ‘easily adapted’ could be misinterpreted and apologize for confusion. In essence, what we were implying earlier in the first rebuttal was that our formulation can guarantee unlearning under the standard definition of unlearning which requires that all samples on F be assigned equal probabilities to all classes. In order to satisfy this requirement with our formulation, we have to solve our formulation several times with different random fractions of F, as discussed in the earlier rebuttal. However, as stated earlier, random-like predictions are not required for us (Remark 2.1).
> > >
> > > Based on our current revision related to this point (Section 2.1, Remark 2.1, final sentence in Discussion), we kindly ask that you can check and let us know if any other modifications could be made to enhance the clarity of our messaging in relation to this point.
> > >
> > > Q9: Figure 2 resolution deterioration.
> > >
> > > We have increased the resolution on our end. Please let us know if it is still deteriorated. Thank you for catching this!

---

> > > > ### Comment · Reviewer_px6p · 2025-10-24
> > > > **Concerns resolved.**
> > > >
> > > > I thank the authors for again providing detailed explanations regarding my open questions! I believe I now understand the intended maintenance scenario and its potential in applications, which resolves my previous concerns.
> > > >
> > > > I adjusted my decision to acknowledge sufficient and clear evidence, and a target audience.

---

> ### Author Response · Authors · 2025-10-23
> **Global comment, continuous learning, targeted unlearning vs model maintenance, and abstract**
>
> We thank the reviewer for another round of helpful and insightful comments regarding our submission. In the resubmitted paper, all new changes are marked in red text (these will be changed to black for a camera ready version).
>
> Q1: Is continual learning a valid option for updating models in scenarios such as new camera models in clinical settings?
>
> Continual learning and unlearning address complementary objectives. Continual learning focuses on adding new information (i.e., accumulate knowledge sequentially) while preserving existing performance on prior data distributions (e.g., preventing catastrophic forgetting) [Wang et al., IEEE TPAMI 2023]. By contrast, our goal is to selectively remove information (or knowledge) from a trained model when data become obsolete, erroneous, or restricted. In scenarios such as decommissioned imaging devices or reclassified diagnostic categories, continual learning, to our knowledge, does not provide a mechanism to selectively discard outdated signals. Hence, while continual learning can accommodate updating a model with the incorporation of new devices, it cannot enforce removal of legacy patterns that may bias subsequent predictions.
>
> Q2: If targeted unlearning equates to a reduction in accuracy on selected samples, how does this have relation to model maintenance in real world clinical settings?
>
> We appreciate the reviewer’s clear understanding that, in our formulation, targeted unlearning induces a deliberate reduction in accuracy on the designated subset (the Forget set). However, in addition to targeted forgetting of individual samples, we also aimed to show that by reshaping the decision boundary after unlearning the behavior of the model on future samples is modified. In other words, after unlearning, unseen samples having the same characteristic (i.e. camera model) of the original forget set should be ‘ignored’ by the model in the future, while performance on the data not marked by the characteristics in the original forget set should be preserved. This is what we describe as model maintenance, or updating a learned model via unlearning based on new real world criteria (see Q4 for a further discussion on scenarios where this may be applicable). In Section 4.2, we report experiments probing unlearning as a tool for maintenance:
>
> -Device unlearning: removing the influence of images acquired from a specific, outdated fundus camera model, and
>
> -Phenotypic unlearning: suppressing the model’s reliance on anatomic outliers such as unusually large vertical palpebral fissures (VPF) in external-eye photographs.
>
> In both cases, the unlearning procedure was applied to training samples meeting the designated criterion (e.g., specific camera type or VPF threshold). The model was then evaluated on a hold-out test set containing both unseen samples that met the same condition as well as samples not meeting the condition. As such, in these experiments F-Acc quantifies accuracy on unseen test samples sharing the forgotten property which is now made much more clear in the revised manuscript. A successful outcome (ie, reduced accuracy on these held-out subsets while preserving performance on the remainder) indicates that the model has removed the corresponding feature dependency from its decision boundary.
>
> We have made these motivations and experimental details more explicit in the revised manuscript (Section 4.2 and Discussion) to clarify how targeted unlearning can serve practical maintenance needs. See Q4 for a more detailed discussion on potential use cases of unlearning in clinical scenarios.
>
> Q3: Overstatement of claims in the abstract
>
> We agree that the original claims in the abstract were too broad, and we have revised the language in the resubmitted version.

---

### Review · Reviewer_PDw9 · 2025-09-17

**Summary Of Contributions:**

The authors propose an unlearning method for neural networks on supervised classification tasks. The presented approach generalizes an existing boundary unlearning method where the model is trained to classify to-be-unlearned samples or classes as the nearest neighboring class instead. (As a result the decision boundary shifts and the unlearned classes are getting subsumed by their neighboring classes). The authors propose a general bilevel optimization formalization for solving the corresponding boundary unlearning problem, with the general objective to misclassify samples of a forget set, while correctly classifying samples of a remain set.

Through the new formalization the authors are able to relate their method to several adjacent frameworks: the initial boundary crossing ("boundary shrinking"), student–teacher approaches, and gradient based adversarial attack methods. Leveraging their more relaxed formalization on inferring boundary conditions, they are able to prove several convergence guarantees for the inner and outer optimization problem of finding the nearest class and retraining the classifier.

Experiments are conducted on CIFAR-10, FashionMNIST and four medical datasets. While I'm not particularly familiar with with standard benchmarks in this field, the evaluation on medical datasets seems to be an important step towards evaluations on more realistic datasets. Results are compared against several seemingly common baselines of the field. Experiments include forgetting of a single class with increasing percentages of sample sizes, and the unlearning of specific samples. Several ablations, a qualitative comparison and inspection of the decision boundary shifts are performed. All experiments seem to be well conducted, provide a good overview of the method's capabilities and indicate a reasonable performance of the model. An additional 'compositional strategy' experiment of their method yielded the overall best performing model on the simple CIFAR-10 dataset. Some concerns on result interpretation/presentation are discussed below.

**Additional Comments:**

-

**Audience:**

Yes

**Audience Explanation:**

The paper relates different methods and aspects of boundary-based unlearning via a unifying formalization. The authors present a broad range of experiments and ablations including real-world datasets. Researchers of the respective field might find the presented results convenient for transporting ideas into the related fields and apply their method on the constructed medical datasets.

**Broader Impact Concerns:**

Given the advertisement of the method for the applications on medical data and privacy-based applications in combination with observed MIA values, I would like to recommend to the claims about unlearning, as discussed in the 'Unlearning' part of the previous 'Claims' section.

**Claims And Evidence:**

No

**Claims Explanation:**

The general formalization and proofs are well presented and seem to be sound. The experimental setup, compared methods and applied metrics seem reasonable and well conducted. However, the obtained results and their interpretation in the text seem to diverge. Privacy guarantees might not hold to the degree promised.



**Interpretation and Presentation of Results**

1) Considering results across the paper (Fig. 2, tables 6-11) it seems that the proposed method features a consistently better forget accuracy, but lower retain accuracy. Throughout the paper, remain set accuracy is stated with "higher is better". However, in Table 1, for both datasets, the author's own method is marked best without being highest (by a large margin of 19% in the first dataset). While the authors focus on the better values in forget accuracy, they fail to critically assess the weakness of their method in terms of remain accuracy. Overall, the results do not reflect to the initial claims of a "exact [and distributional] unlearning", as the proposed method seems to consistently forget larger portions of the remain set in comparison to the other baselines.
2) Table 1 states that the MIA metric is the membership inference attack accuracy. I, therefore, assume that a lower value is better? (Also the "Higher/lower is better" indicator for MIA metric in Fig. 2 is missing.) However, the NegGrad method with the lowest value of 0.3 (vs. 0.5 which is neither the lowest nor highest, with several other unmarked 0.5 entries existing) is not indicated as best.
3) All R Acc and F Acc values are reported without standard errors, as only the best model seems to be selected from all seeds for testing. The lack of error bounds weakens the reliability of the results.



**Unlearning**

Given, that the underlying Chen et al. method shifts unlearned samples to the decision boundaries of the classes, it is likely that such 'unlearned' samples might again be found and extracted by particularly searching for logit configurations near the decision boundaries. I am therefore critical of the author's claims of the method to consistently "unlearn" and its ability to perform "instance-level" model maintenance. As samples are relabeled, but might not be forgotten, single sample memorization might still occur. This assumption of incomplete unlearning seems to be supported by the consistently high MIA values -- in comparison to other baseline methods (except for the baseline Boundary method, which suffers the same problem) as for example reported in Figure 1/Table 1. Given, that the authors advertise their method for the particularly privacy sensible domain of medical data, I would like to recommend the authors to revise their claims and better highlight the limitations of the approach in this regard.

**Requested Changes:**

My requested changes primarily concern on the interpretation and presentation of experimental results, as well as the unlearning claims as discussed in the previous fields. I consider both, the presentation of results and unlearning claims critical, as they might lead to an inaccurate perception of the method's capabilities.



1) Considering the inner minimization constraint in Eq. (1), it is not directly obvious to me, why choosing $\kappa = 1/K$ projects samples directly onto the decision boundary. If I'm not mistaken, the particular choice of 1/K being relating to the exact boundary is never derived in the paper or appears in the appendix. Considering Sec. 2.3 it seems that the particular choice of $\kappa = 1/K$ might guarantee a switch in the label as some other class is guaranteed to yield a higher value. This however might not correspond to the actual decision boundary. I would kindly like to ask the authors to clarify.



Recommendations for minor adjustments:

* page 2, last paragraph: "... extend the boundary-based unlearning framework of Chen et al. (Chen et al. 2023)". Use citet to avoid the repeated "Chen et al.".
* Fig. 8 and Table 6 are scaled too large. Remove the stretch to line width.
* Start of page 3: "Please see Appendix [reference missing?] for more details."
* Table 6: The NegGrad entry lacks F acc blue coloring for fashionMNIST and overall red indicators. (Also: typo "fashioMNIST").
* MIA results are never color coded/bolded in the appendix.
* Tables 1,2,4-11: I would like to suggest making the lines of the tables connected if possible (similar to table 3) and/or remove some of the rules for cleaner tables.

---

> ### Author Response · Authors · 2025-10-09
> **Response to concerns regarding the interpretation and presentation of results**
>
> Q: The proposed method has good accuracy on F with relatively poorer performance on R accuracy. The authors need to critically assess this tradeoff in the manuscript.
>
>
> This is an important detail of our manuscript that we are grateful to address. While other methods do have slightly higher accuracy on R at the various forget set sizes (percentages) compared to our method (as well as the original boundary shifting method proposed by Chen et. al.), this, in many cases comes at the cost of poor forgetting which is undesirable. The F/R ratio metric summarizes this tradeoff (where lower is better).
>
> We agree with the reviewer that our results section would benefit from a more nuanced discussion of this tradeoff.
>
> Revision: We have updated our manuscript to explicitly acknowledge this tradeoff in R-Acc in Section 4.1, and we have adjusted the introduction to avoid overclaiming exact and distributional unlearning
>
> Q: In Table 1, for both datasets, the author's own method is marked best without being highest (by a large margin of 19% in the first dataset).
>
> We thank the reviewer for pointing this out and apologize for any confusion this has caused. We have marked all values in bold that satisfy the following criterion: in this table, we only consider all values such that F/R< 0.5, and among this subset, the ones with lowest F and highest R are marked in bold. We believe that this would allow comparison between results that had some amount of forgetting.
>
> Revision: We have rectified this to bold now the highest R and lowest F irrespective of whether forgetting occurred or not.  We have also added the above criterion in the legend.
>
>
> Q: Are lower MIA values the best? The current tables are unclear as to what the top best MIA value is.
>
>
> We followed the protocol of Kurmanji et al. (2024). As described in Section 3.3, MIA accuracy is evaluated with a logistic regression attack on clipped loss values. A perfect defense achieves 50% accuracy, i.e., indistinguishability. Thus, values closer to 0.50 are better, not lower in absolute terms.
>
> Revision: We have corrected all figure captions and table legends (including Fig. 2 and all relevant Tables) to explicitly note that values closer to 0.50 are better, and we now bold the entries closest to 0.50 for clarity.
>
>
>
> Q: All R Acc and F Acc values are reported without standard errors, as only the best model seems to be selected from all seeds for testing. The lack of error bounds weakens the reliability of the results.
>
> We agree that reporting variance would strengthen the presentation. We would like to emphasize that the trends we report, particularly the tradeoff between forgetting (F-Acc) and retention (R-Acc), are consistent across datasets and baselines, and we do not expect conclusions to change with multi-seed reporting. We will add error bars in the final revision as it takes considerable computational time to repeat all of the experiments.

---

> ### Author Response · Authors · 2025-10-09
> **Response to concerns regarding unlearning claims**
>
> Q: As samples are moved to the boundary of the decision space, is it possible that unlearned samples can be reidentified by searching for specific configurations of logits? As samples are relabeled, but might not be forgotten, single sample memorization might still occur.
>
> On the one hand, it is unlikely to identify samples that were unlearned based on its logits alone. For instance, if the learning problem is hard (i.e., many samples near the decision boundary), then searching for logits alone will not be very informative. In more detail, if we consider the logits of a point that was unlearned alone (by shifting the decision boundary) as well as a point that was correctly classified (ie, has the same label that was used to forget the original point) but near the decision boundary, the logit configuration of these points are likely similar. We have modified Figure 1 with additional purple points near the decision boundary to illustrate.
>
> On the other hand, if the forget set had points close to each other and all of them have the same (nearest) adversarial label, then by searching logits, it may be possible to identify them. In recent works, such as Hu et. al. ICLR 2025, if the “attacker” has some information from the forget set, then it may be possible to identify points that were unlearned. The reviewer will note that the use cases we consider in medical settings, in-house datasets are HIPAA protected and so the risks of information leakage are minimal. We will have clarified this in the limitation section and added the noted citation.
>
> Furthermore, using logit configurations to reidentify unlearned points is one of the driving ideas behind MIAs. We note that on clinical datasets, our method achieved competitive MIA values (0.58 and 0.70 in Table 1), indicating that it performs reasonably well even outside its intended scope of model maintenance. Please see our earlier response to for a further discussion on ideal MIA values, in particular, higher MIA values are not necessarily bad.
>
> In our revised manuscript, we have clarified that our intended use case is not strictly privacy, but rather model maintenance in medical settings. We have also added a more detailed interpretation of MIA values in the discussion, and further softened the claims related to MIA as a metric in Section 3.3.
>
> Q: Given the sensitive nature of medical data, the manuscript would benefit from a discussion on the limitations of unlearning in medical settings.
>
> We fully agree that unlearning in healthcare settings raises serious ethical considerations. In response to reviewer feedback (yourself as well as px6p), we have added a new section, Implications of Unlearning in Healthcare, where we explicitly caution against inappropriate deployment of unlearned models in diagnostic contexts and highlight the need for registries, governance, and regulatory oversight. We hope these clarifications make clear both the methodological contribution of our work and its limitations with respect to privacy guarantees.

---

> ### Author Response · Authors · 2025-10-09
> **Clarification on the bound $\kappa = 1/K$**
>
> We will explain our rationale for using $\kappa = 1/K$.
>
> Formally, any classification model outputs a probability vector $p(x)$ with entries $\{p_j(x)\}_{j=1}^K$ that sum to 1. A sample is classified to class $y$ if $p_y(x) > p_j(x)$ for all $j\neq y$. Therefore, the \textit{decision boundary} for class $y$ is simply the set of $x$ such that there exists some $j\neq y$ with $p_y(x)=p_j(x)$.
>
> The threshold $\kappa=1/K$ provides the tightest uniform condition under which class $y$ can no longer be strictly dominant:
>
>
>  - If $p_y(x) > 1/K$, class $y$ may still be the maximizer.
>  - If $p_y(x) = 1/K$, then at least one other class satisfies $p_j(x)\geq 1/K$. Two situations can occur:
>         1. If some $j$ has $p_j(x) > 1/K$, then $y$ cannot be the class (the point x lies across the boundary).
>         2. If all classes equal $1/K$, then $y$ is tied with others, which by definition is a boundary case.
>   - If $p_y(x) < 1/K$, then $y$ is dominated by another class, meaning the point x is already across the boundary or from other x with y as their label.
>
> Thus, $\kappa=1/K$ does not guarantee an immediate label switch, but it does guarantee that the perturbed point is no longer in the \textit{interior} of class $y$’s region. In other words, it lies either exactly on a decision boundary (via a tie) or across it (via another class overtaking). This makes $\kappa=1/K$ the natural threshold for our boundary search procedure.
>
> We will clarify this reasoning explicitly in the appendix of the revised manuscript.

---

> ### Author Response · Authors · 2025-10-09
> **Minor requested adjustments**
>
> Q: page 2, last paragraph: "... extend the boundary-based unlearning framework of Chen et al. (Chen et al. 2023)". Use citet to avoid the repeated "Chen et al.".
>
> Revision: Modified for easier reading. Thank you!
>
>
> Q: Fig. 8 and Table 6 are scaled too large. Remove the stretch to line width.
>
> Revision: We have resized both figures to .8 of line width.
>
> Start of page 3: "Please see Appendix [reference missing?] for more details."
>
>
> Q: We will fix this reference link, but we do not see the missing reference in the document. Could you please point out or be specific about where you saw this?
>
> Q: Table 6: The NegGrad entry lacks F acc blue coloring for fashionMNIST and overall red indicators. (Also: typo "fashioMNIST").
>
> Revision: We have corrected this error in  Figure 6 - thank you for highlighting it!
>
> Q: MIA results are never color coded/bolded in the appendix.
>
> Revision: We have marked the MIA values closest to .5 in bold in all tables in the appendix and updated the figure legends to denote that this is the ideal score in MIA testing.
>
> Q: Tables 1,2,4-11: I would like to suggest making the lines of the tables connected if possible (similar to table 3) and/or remove some of the rules for cleaner tables.
>
> We will reformat all tables in the final revision.

---

> > ### Comment · Reviewer_PDw9 · 2025-10-16
> >
> > I appreciate the author's comprehensive rebuttal, including the added explanations in sections 2 which clarify their goals in terms of unlearning and forgetting for model maintenance, focusing on the mitigation of apparent effects of set of samples for, possibly non-public, models. This offsets the intended semantics of these terms from these of privacy-driven domains. The added "Implications of Unlearning in Healthcare" section now clearly outlines possible risks and highlights the need for safeguards in critical environments. Similarly, I appreciate the further remarks on the interpretation of $\kappa$ and the adjustments to the presentation and interpretation of presented results.
> >
> > While I will still follow the ongoing discussions with the other reviewers, my initial points have been resolved.
> >
> >
> > Regarding, the "Please see Appendix for more details.", I was referring to the specific sentence at the end of Section 2.2 (following Lemma 2.2). As all other mentions of additional materials in the paper always linked to specific appendices, I was wondering whether the particular sentence referred to some additional appendix that was simply not linked or to the aforementioned appendix A.1 in the paragraph. Given your answer, I now assume the latter.

---

> > > ### Author Response · Authors · 2025-10-24
> > > **Thank you for your comments**
> > >
> > > `Given your answer, I now assume the latter.`
> > >
> > > This is the correct interpretation!
> > >
> > > All of the authors kindly thank you for the time and care spent reviewing our manuscript, as feedback from your review has greatly improved the work.

---

### Review · Reviewer_zNwu · 2025-09-26

**Summary Of Contributions:**

#### Summary of Contributions
The paper introduces a novel approach to machine unlearning by recasting it as a bilevel optimization problem. It proposes an algorithm that uses perturbed sign-gradient methods to identify decision boundary points for relabeling forget samples, enabling targeted forgetting while preserving performance on remaining data. The method includes convergence guarantees for the inner optimization loop and supports tunable loss designs to control the forgetting-retention tradeoff. Additionally, it allows for compositional merging of unlearned models. Empirical evaluations on benchmark datasets (CIFAR-10, FashionMNIST) and clinical medical imaging datasets (MRI, color fundus photographs, oculoplastic images) demonstrate superior performance over baselines in terms of forgetting efficacy, retention accuracy, and resistance to membership inference attacks.

#### Key Strengths
- Provides a principled bilevel optimization framework with theoretical convergence guarantees, extending prior boundary-based methods.
- Applies unlearning to practical clinical scenarios, such as removing influence from deprecated imaging devices or anatomical outliers, which is underexplored.
- Offers tunability via hyperparameters and supports model composition, enhancing flexibility for real-world model maintenance.
- Comprehensive experiments across diverse datasets, including ablation studies, validate the method's effectiveness and efficiency compared to retraining and other unlearning techniques.

#### Key Weaknesses
- Relies on hyperparameter tuning (e.g., γ, λ, ϕ), which may require additional computational effort or expertise.
- While faster than full retraining, the method's computational cost could be prohibitive for very large-scale models or datasets.

**Audience:**

Yes

**Audience Explanation:**

Machine unlearning is a growing area in machine learning, particularly for privacy preservation, model updating, and compliance with regulations like GDPR. The paper's focus on clinical applications addresses real-world challenges in healthcare ML, where data shifts and device changes are common. The theoretical guarantees, tunability, and compositional aspects would appeal to researchers in optimization, adversarial robustness, and medical AI. Findings on balancing forgetting and retention could inform practical deployments, making it relevant to TMLR's broad audience in trustworthy and applied ML.

**Broader Impact Concerns:**

None.

**Claims And Evidence:**

Yes

**Claims Explanation:**

The theoretical claims, such as convergence of the perturbed sign-gradient method, are supported by proofs in the appendices, including applications of KKT conditions and stochastic optimization arguments. Empirical claims are backed by extensive experiments on six datasets, with quantitative metrics (F-Acc, R-Acc, MIA accuracy) reported in tables and figures. Comparisons to baselines like Boundary Shrink, SCRUB, and retraining show consistent improvements. Ablation studies isolate the impact of key components, and timing analyses confirm efficiency. Visualizations and qualitative examples further strengthen the evidence. Overall, the results are reproducible based on described hyperparameters and setups.

**Requested Changes:**

1. Include more details on hyperparameter selection, such as the grid search ranges used and sensitivity analyses beyond the ablations, to aid reproducibility. (Strengthening: Enhances practical utility.)

2. Provide experiments on additional modalities or larger-scale models (e.g., transformer-based architectures) to broaden applicability.

3. In Section 4.1, the described forget set percentages are 1%, 10%, 25%, 50%, and 75%, but Figure 2 includes a 100% data point, which is not mentioned in the setup; this discrepancy requires explanation.

4. Discuss any specific adaptations made to the algorithm for medical images (e.g., handling large sizes or sparse regions) or explain why the general method is sufficient for these datasets.

5. Elaborate in the main text on how the new label y_b is initially assigned (e.g., via argmax on the boundary point) and provide concrete examples of how to modify it for directing forgetting to specific classes.

No critical changes are needed, as the current submission is already strong in methodology, evaluation, and novelty.

---

> ### Author Response · Authors · 2025-10-09
> **Response to all requested changes made by the reviewer**
>
> Q: Include more details on hyperparameter selection, such as the grid search ranges used and sensitivity analyses beyond the ablations, to aid reproducibility.
>
> In our experiments, we conducted grid searches over $\lambda \in {0,10^{-4},10^{-3},10^{-2},10^{-1}}$ and $\gamma \in {0,10^{-4},10^{-1},1}$, as specified in Section 4.1. The results of these searches, including sensitivity heatmaps, are provided in the Appendix (Figures 10,11,12). For each dataset, the best-performing pair $(\gamma,\lambda)$ was selected for reporting in the main tables.
>
> Revision: We have revised Section 4.1 of the manuscript to make these ranges and procedures more explicit.
>
> Q: Provide experiments on additional modalities or larger-scale models (e.g., transformer-based architectures) to broaden applicability.
>
> While we are very excited about the future direction of this research and studying unlearning in larger clinical models, our current approach is not adapted towards transformers or other non-CNN architectures. In the final paragraph of our discussion, we note that extending our methods to attention based architectures as well as larger models (e.g. foundational models) is a key direction to pursue in the future. We will include this discussion in the final revision.
>
> Q: In Section 4.1, the described forget set percentages are 1%, 10%, 25%, 50%, and 75%, but Figure 2 includes a 100% data point, which is not mentioned in the setup; this discrepancy requires explanation.
>
> Revision: We have modified the first paragraph of section 4.1 to include 100% as a forget set percentage. The results for unlearning an entire class are discussed in paragraph 4 of section 4.1. We agree that adding it explicitly as a condition early in the section strengthens the setup.
>
>
> Q: Discuss any specific adaptations made to the algorithm for medical images (e.g., handling large sizes or sparse regions) or explain why the general method is sufficient for these datasets.
>
> Our unlearning algorithm does not require modality-specific adaptations, as it operates on model outputs and loss functions rather than raw image structure. Standard preprocessing sufficed for all medical datasets, making the procedure identical to that used on benchmark datasets.
>
> Revision: We have clarified this in Section 3.1.
>
> Q: Elaborate in the main text on how the new label y_b is initially assigned (e.g., via argmax on the boundary point) and provide concrete examples of how to modify it for directing forgetting to specific classes.
>
> We thank the reviewer for this helpful suggestion.
>
> Revision: In the revised manuscript (Section 2.4), we now explicitly state that $y_i^b$ is assigned as the argmax of the logits at the boundary point, i.e., the highest scoring non-true class. We also clarified that $y_i^b$ need not be restricted to this choice; it can alternatively be set to a fixed target class, thereby steering forgetting toward that class.

---

### Author Response · Authors · 2025-10-09
**General Response**

We thank all reviewers for their thoughtful and constructive feedback. The primary conceptual clarification raised by Reviewer px6p concerned the definition of unlearning and its desiderata. We have addressed this by explicitly contrasting our operational definition, misclassification-based forgetting for model maintenance, with the retraining-based paradigm standard in privacy-driven unlearning (Section 2.1, Remark 2.1). The remainder of the feedback focused on experimental clarity, metric interpretation, and presentation details. In the revised manuscript, we have refined figure legends and table formatting, added missing methodological details (e.g., hyperparameter ranges, label assignment, inclusion of 100% forget-set condition), clarified the interpretation of MIA values, and added a new discussion section on the ethical implications of unlearning in healthcare (Section 6). We believe these revisions have substantially improved the clarity, rigor, and framing of the work while incorporating all reviewer suggestions.

We have uploaded the revised manuscript for your review. All changes are highlighted in blue text which will be converted to black in the camera ready version.

---

### Decision · Action_Editor_nFN2 · 2025-11-02

**Recommendation:** Accept as is

**Audience:**

Yes

**Audience Explanation:**

Practically important problem.

**Claims And Evidence:**

Yes

**Claims Explanation:**

The proposal is effective, demonstrated by experiments on many datasets (including standard natural image datasets and medical image datasets as well).

---

> ### Author Response · Authors · 2025-12-01
> **Camera Ready Submitted**
>
> To reviewers,
>
> We have submitted the camera ready version. Once again, thank you to all for a helpful and productive review process which we believe greatly improved the quality of our manuscript.
>
> Sincerely,
> All Authors